



Atmospheric
Chemistry
and Physics

# The production and hydrolysis of organic nitrates from OH radical oxidation of $\beta$-ocimene

**Ana C. Morales**[1,★], **Thilina Jayarathne**[1,a,★], **Jonathan H. Slade**[2], **Alexander Laskin**[1,3], **and Paul B. Shepson**[1,3,4]

[1]Department of Chemistry, Purdue University, West Lafayette, IN 47906, USA
[2]Department of Chemistry & Biochemistry, University of California San Diego, La Jolla, CA 92093, USA
[3]Department of Earth, Atmospheric, and Planetary Sciences, Purdue University, West Lafayette, IN 47906, USA
[4]School of Marine and Atmospheric Sciences, Stony Brook University, Stony Brook, NY 11794, USA
[a]now at: Bristol Myers Squibb, 1 Squibb Drive, New Brunswick, NJ 08901, USA
★These authors contributed equally to this work.

**Correspondence:** Paul B. Shepson (paul.shepson@stonybrook.edu)

**Abstract.** Biogenic volatile organic compounds (BVOCs) emitted by plants represent the largest source of non-methane hydrocarbon emissions on Earth. Photochemical oxidation of BVOCs represents a significant pathway in the production of secondary organic aerosol (SOA), affecting Earth's radiative balance. Organic nitrates ($RONO_2$), formed from the oxidation of BVOCs in the presence of $NO_x$, represent important aerosol precursors and affect the oxidative capacity of the atmosphere, in part by sequestering $NO_x$. In the aerosol phase, $RONO_2$ hydrolyze to form nitric acid and numerous water-soluble products, thus contributing to an increase in aerosol mass. However, only a small number of studies have investigated the production of $RONO_2$ from OH oxidation of terpenes, and among those, few have studied their hydrolysis. Here, we report a laboratory study of OH-initiated oxidation of $\beta$-ocimene, an acyclic, tri-olefinic monoterpene released during the daytime from vegetation, including forests, agricultural landscapes, and grasslands. We conducted studies of the OH oxidation of $\beta$-ocimene in the presence of $NO_x$ using a 5.5 m$^3$ all-Teflon photochemical reaction chamber, during which we quantified the total (gas- and particle-phase) $RONO_2$ yield and the SOA yields. We sampled the organic nitrates produced and measured their hydrolysis rate constants across a range of atmospherically relevant pH. The total organic nitrate yield was determined to be $38(\pm 9)\%$, consistent with the available literature regarding the dependence of organic nitrate production (from $RO_2$ + NO) on carbon number. We found the hydrolysis rate constants to be highly pH dependent, with a hydrolysis lifetime of $51(\pm 13)$ min at pH = 4 and $24(\pm 3)$ min at pH = 2.5, a typical pH for deliquesced aerosols. We also employed high-resolution mass spectrometry for preliminary product identification. The results indicate that the ocimene SOA yield ($< 1\%$) under relevant aerosol mass loadings in the atmosphere is significantly lower than reported yields from cyclic terpenes, such as $\alpha$-pinene, likely due to alkoxy radical decomposition and formation of smaller, higher-volatility products. This is also consistent with the observed lower particle-phase organic nitrate yields of $\beta$-ocimene – i.e., $1.5(\pm 0.5)\%$ – under dry conditions. We observed the expected hydroxy nitrates by chemical ionization mass spectrometry (CIMS) and some secondary production of the dihydroxy dinitrates, likely produced by oxidation of the first-generation hydroxy nitrates. Lower $RONO_2$ yields were observed under high relative humidity (RH) conditions, indicating the importance of aerosol-phase $RONO_2$ hydrolysis under ambient RH. This study provides insight into the formation and fate of organic nitrates, $\beta$-ocimene SOA yields, and $NO_x$ cycling in forested environments from daytime monoterpenes not currently included in atmospheric models.

**Published by Copernicus Publications on behalf of the European Geosciences Union.**

# 1 Introduction

Biogenic volatile organic compounds (BVOCs) constitute the largest flux (88 %) of all non-methane organic compounds to the atmosphere (Goldstein and Galbally, 2007; Guenther et al., 1995, 2012). Isoprene and monoterpenes account for $\sim 60\,\%$ of the total global BVOC emissions (Goldstein and Galbally, 2007; Guenther et al., 1995). BVOCs participate in chemical reactions with regulated air pollutants, including $NO_2$ and $O_3$, and with radical species such as OH and $NO_3$, which leads to the formation of low-volatility oxygenated compounds that partition into aerosol particles and represents a source of secondary organic aerosol (SOA) (Atkinson and Arey, 2003; Hallquist et al., 2009; Hatakeyama et al., 1991; Isaksen et al., 2009; Lee et al., 2016; Monks et al., 2009; Perring et al., 2013; Pye et al., 2015; Tuazon and Atkinson, 1990). Globally, the oxidation of BVOCs emitted from forests represents the largest source of SOA and affects climate and air quality (Hallquist et al., 2009).

One specific group of BVOC oxidation products of interest is organic nitrates ($RONO_2$). By sequestering $NO_x$, $RONO_2$ slow the production of tropospheric $O_3$, but it can act as a source of $NO_x$ and $O_3$ downwind upon further oxidation and decomposition (Browne and Cohen, 2012; Pusede et al., 2015). These low-volatility, water-soluble compounds are important precursors to and constituents of SOA due to their efficient partitioning into the condensed phase (Biesenthal et al., 1997; Fry et al., 2009, 2014; Perraud et al., 2012; Rollins et al., 2012). Ambient measurements have indicated that up to 23 % of molecules in organic aerosol contain the $RONO_2$ functional group, suggesting $RONO_2$ represent a significant fraction of BVOC-derived oxidation products (Ditto et al., 2020; Rollins et al., 2013). However, the hydrolysis lifetime of organic nitrates in the aerosol phase ($\sim 1$–$3\,h$) is much shorter than the lifetime of the aerosol itself to deposition ($\sim 1$ week), which suggests the contribution of organic nitrates to aerosol mass may have been underestimated from the observational data (Romer et al., 2016).

It has been shown that these low-volatility oxidation products readily partition from the gas phase into the particle phase and undergo further reactions (Perraud et al., 2012). Once in the particle phase, organic nitrates can undergo rapid hydrolysis under acid-catalyzed conditions, representing a sink for $NO_x$ in the form of $HNO_3$ (Bean and Hildebrandt Ruiz, 2016; Darer et al., 2011; Hu et al., 2011; Liu et al., 2012; Rindelaub et al., 2015; Takeuchi and Ng, 2019). Hydrolysis rates and the relative contribution of particle-phase hydrolysis as a sink for organic nitrates (compared to reaction with $O_3$, OH, photolysis, and deposition) vary significantly and depend on molecular structure (Boyd et al., 2015; Zare et al., 2018). To date, there are few studies of hydrolysis rate constants for monoterpene-derived organic nitrates (Rindelaub et al., 2015). The chemical structure (primary, secondary, and tertiary) can influence the rate of hydroly-

sis lifetimes of organic nitrates (Bean and Hildebrandt Ruiz, 2016; Boyd et al., 2015, 2015, 2017; Darer et al., 2011; Hu et al., 2011; Liu et al., 2012; Ng et al., 2017; Rindelaub et al., 2016b; Takeuchi and Ng, 2019; Zare et al., 2018), which is sometimes rapid compared to the lifetime of the particles with respect to dry deposition. Thus, more studies are needed to assess the structural dependence of both $RONO_2$ production yields and the hydrolysis rate constants.

Although monoterpenes contribute significantly to the total annual BVOC budget, current models underestimate the impact of $RONO_2$ on SOA mass measured in an ambient environment (Pye et al., 2013). Importantly, $RONO_2$ yields for many BVOCs are unknown. Monoterpenes have been shown to produce more SOA than from isoprene in boreal regions (Lee et al., 2006; Tsigaridis and Kanakidou, 2007). Some of that SOA production from terpenes is now believed to be due to organic peroxides produced from $RO_2 + RO_2$ reactions involving the 10-carbon $RO_2$ radicals when NO concentrations are low (Heinritzi et al., 2020). Pratt et al. (2012) showed that, for a temperate mixed forest in Michigan, isoprene-derived nitrates dominate the daytime simulated gas-phase organic nitrates; however, $\sim 20\,\%$ is derived from monoterpene nitrates. Of this, nearly one-third results from OH-initiated oxidation of $\beta$-ocimene, a tri-olefinic monoterpene of high reactivity that is emitted during the daytime (Pratt et al., 2012). To the best of our knowledge, the production of $\beta$-ocimene organic nitrates and aqueous-phase processing of its atmospheric oxidation products have not previously been studied.

To expand our understanding of the chemical evolution and fate of monoterpene-derived $RONO_2$, we investigated the formation of $RONO_2$ produced from the OH-initiated oxidation of $\beta$-ocimene in the presence of $NO_x$ using a photochemical reaction chamber. These experiments were conducted as a function of chamber relative humidity (RH) to provide insight on aqueous-phase partitioning and processing. The aqueous-phase processing of these species was explored using linear quadrupole ion trap mass spectrometry to determine hydrolysis kinetics. This study aims to quantify the yields of $\beta$-ocimene-derived $RONO_2$ and the hydrolysis rate constants to better understand how organic nitrates may impact air quality and climate.

# 2 Methods

Experiments were conducted in a custom-made 5.5 m³ photochemical reaction chamber consisting of perfluoroalkoxy (PFA)–Teflon walls, PFA-coated endplates, a PFA-coated mixing fan, and UV lamps as described in Chen et al. (1998) and Lockwood et al. (2010). These experiments were conducted as a function of chamber RH, keeping other variables as constant as possible, and at 24($\pm$2) °C. Eighteen experiments were carried out under variable humidity conditions, i.e., $< 3\,\%$ up to 70 % RH, as listed in Table 1. The maxi-

mum humidity achievable in this chamber is 70 %, so higher RH is not considered for this study.

Before the experiment, the chamber walls were cleaned by adding $\sim 500$ ppb $O_3$, irradiating for 1 h, and flushing with ultra-zero (UZ) air in the dark until the $O_3$ concentration was $\sim 0$ ppb. The $O_3$ concentration was monitored using a dual-beam ozone monitor (2B Technologies, model 205), which is calibrated using an ozone calibration source (2B technologies, model 306). The chamber RH was monitored using a LI-COR hygrometer (model 7000). The LI-COR was calibrated from known RH air sampled over a saturated $K_2SO_4$ solution at a fixed temperature. Experiments were started only if the chamber particle number concentration was $< 10$ particles per cubic centimeter.

First, $\beta$-ocimene ($\geq 90$ %, Sigma-Aldrich; (E)-/(Z)-$\beta$-ocimene mixture) was introduced to the chamber via injection through a T-shaped heated pyrex inlet, through which ultrapure $N_2$ was used to transfer the injected $\beta$-ocimene via 1/4 in. PFA tubing at a flow rate of 5 L min$^{-1}$. Formaldehyde (37 % $v/v$ in water, Sigma) was injected using the same setup, followed by pure NO (99.5 %, Praxair) injected using the same setup without heat. Formaldehyde photolysis serves as an OH precursor in the presence of NO (Possanzini et al., 2002). Ammonium sulfate (($NH_4$)$_2SO_4$) seed particles were generated using a 10 wt % ($NH_4$)$_2SO_4$ aqueous solution and a commercial atomizer (TSI, Inc., model 3076), and subsequently dried by passing through a diffusion dryer prior to entering the reaction chamber (as in Rindelaub et al., 2015). For humid experiments, chamber RH was adjusted by bubbling UZ air through nano-pure water using a commercial bubbler immersed in a temperature-controlled bath. The chamber fan was started during the initial injections, ensuring reactants were well mixed. Following mixing, initial concentrations were measured before the chamber UV lights were activated. After initial concentrations were measured, the fan was stopped to minimize wall losses, the chamber lights were activated (time = 0), and real-time measurements were obtained. A representative experimental time series is shown in Fig. 1, where the area between the blue vertical lines represents the time when the UV lights were switched on.

The $\beta$-ocimene concentration was quantified using gas chromatography–flame ionization detection (GC-FID; HP 5890 Series II) equipped with a gas injection loop. The GC-FID was calibrated using the same $\beta$-ocimene standard injected during the experiments, with gas-phase concentrations prepared in $\sim 200$ L PFA–Teflon bags. Seven-point calibration curves with $R^2 > 0.995$ were obtained and used for $\beta$-ocimene quantitation. NO and $NO_2$ concentrations were measured using a custom-built chemiluminescence $NO_x$ analyzer (Lockwood et al., 2010). The size-resolved particle mass concentration was determined using a scanning mobility particle sizer (SMPS, TSI, Inc., model 3062) directly connected to the chamber via copper sampling lines. The hydroxy nitrates were quantified using I$^-$ chemical ionization

**Table 1.** Experimental conditions and yield summary of individual experiments. Error represents the propagated analytical uncertainty.

| Exp no. | Seed | Irradiation time (min) | RH (%) | Δ [β-Ocimene] (ppb) | Δ [NO] (ppb) | Δ [NO₂] (ppb) | Δ [SOA mass] (μg m⁻³) | Gas phase -ONO₂ yield (%) | Particle phase -ONO₂ yield (%) | Hydroxy nitrate yield (%) |
|---|---|---|---|---|---|---|---|---|---|---|
| 1 | (NH₄)₂SO₄ | 19 | < 3 % | 266 ± 67 | 509 ± 57 | 475 ± 45 | 96 ± 10 | nm | nm | nm |
| 2 | (NH₄)₂SO₄ | 54 | < 3 % | 223 ± 30 | 392 ± 29 | 284 ± 14 | 63 ± 5 | nm | nm | nm |
| 3[a] | (NH₄)₂SO₄ | 82 | < 3 % | 1357 ± 102 | 2604 ± 101 | nm | 933 ± 62 | 24 % ± 6 % | 4.1 % ± 1.1 % | 7(+2/−5) % |
| 4 | (NH₄)₂SO₄ | 73 | < 3 % | 425 ± 58 | 914 ± 13 | 587 ± 24 | nm | 42 % ± 11 % | 1.5 % ± 0.5 % | 6(+2/−4) % |
| 5 | (NH₄)₂SO₄ | 78 | < 3 % | 645 ± 49 | 1448 ± 11 | 817 ± 40 | nm | 38 % ± 11 % | 1.6 % ± 0.4 % | nm |
| 6 | (NH₄)₂SO₄ | 82 | < 3 % | 516 ± 44 | 1540 ± 11 | 852 ± 33 | 143 ± 11 | 40 % ± 11 % | 1.3 % ± 0.3 % | 7(+2/−5) % |
| 7 | (NH₄)₂SO₄ | 102 | < 3 % | 524 ± 48 | 1091 ± 19 | 856 ± 33 | 189 ± 14 | 37 % ± 10 % | 1.3 % ± 0.3 % | 9(+2/−6) % |
| 8 | none | 95 | < 3 % | 653 ± 45 | 1691 ± 11 | 1021 ± 38 | 101 ± 6 | 29 % ± 8 % | 1.5 % ± 0.4 % | 8(+2/−5) % |
| 9 | (NH₄)₂SO₄ | 92 | 46 % | 556 ± 44 | 1545 ± 52 | 1041 ± 38 | 238 ± 17 | 20 % ± 4 % | 1.2 % ± 0.3 % | nm |
| 10 | (NH₄)₂SO₄ | 89 | 49 % | 692 ± 47 | 2019 ± 11 | 1185 ± 43 | 277 ± 24 | 20 % ± 4 % | 0.9 % ± 0.2 % | 7(+2/−5) % |
| 11 | none | 19 | < 3 %–54 %[b] | 666 ± 38 | nm | nm | BDL | 18 % ± 4 % | BDL | 6(+2/−4) % |
| 12 | (NH₄)₂SO₄ | 72 | 51 % | 798 ± 48 | 1913 ± 16 | 1034 ± 39 | 315 ± 23 | 22 % ± 5 % | 1.0 % ± 0.2 % | 3(+2/−2) % |
| 13 | (NH₄)₂SO₄ | 84 | 70 % | 641 ± 46 | 1628 ± 12 | 721 ± 41 | 269 ± 19 | 19 % ± 4 % | 0.2 % ± 0.2 % | 2(+1/−1.3) % |
| 14 | none | 24 | 67 % | 407 ± 56 | 474 ± 11 | 90 ± 30 | BDL | 17 % ± 4 % | 0.8 % ± 0.2 % | nm |
| 15 | (NH₄)₂SO₄ | 92 | 69 % | 783 ± 47 | 1643 ± 10 | 1209 ± 51 | 417 ± 26 | 14 % ± 3 % | 0.8 % ± 0.2 % | nm |
| 16 | (NH₄)₂SO₄ | 86 | 13 % | 648 ± 49 | nm | nm | 238 ± 18 | 26 % ± 6 % | 1.7 % ± 0.5 % | nm |
| 17 | (NH₄)₂SO₄ | 87 | < 3 % | 569 ± 41 | 1418 ± 20 | 878 ± 39 | 231 ± 18 | 34 % ± 7 % | 1.8 % ± 0.5 % | nm |
| 18 | (NH₄)₂SO₄ | 88 | < 3 % | 560 ± 47 | 1256 ± 16 | 666 ± 27 | 156 ± 12 | 35 % ± 8 % | 1.6 % ± 0.4 % | nm |

[a] High-concentration experiment. Not used for yield calculations. [b] Irradiated under dry conditions, then increased RH in the dark up to 54 %. nm – not measured; BDL – below the detection limit.

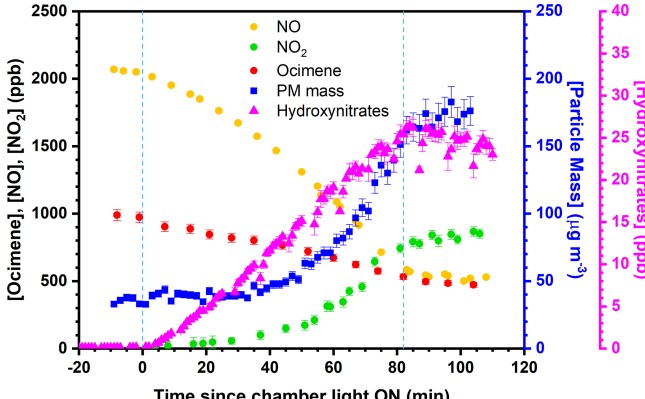

**Figure 1.** Time series of $\beta$-ocimene oxidation and particle formation for experiment no. 6. Measurements presented are representative of all online measurements obtained during chamber experiments. The error bars represent the propagated analytical uncertainty and blue dashed lines represent the time of UV light irradiation. (PM = particulate matter).

mass spectrometry (CIMS) (Xiong et al., 2015, 2016). The inlet of the CIMS was maintained at a high relative humidity via saturated ultra-high-purity $N_2$ carrier flow to eliminate the effects of water vapor on the CIMS sensitivity (Lee et al., 2016; Xiong et al., 2015). A synthesized $\alpha$-pinene hydroxy nitrate standard was used as a surrogate standard to calibrate the CIMS for $\beta$-ocimene hydroxy nitrate determination (Rindelaub et al., 2016b; Slade et al., 2017). We note there may be differences in the sensitivity of the instrument towards individual $\beta$-ocimene hydroxy nitrate isomers compared to the $\alpha$-pinene hydroxy nitrate standard due to differences in polarity and thus binding affinity with $I^-$ (Iyer et al., 2016; Lopez-Hilfiker et al., 2016). The synthetic $\alpha$-pinene hydroxy nitrate standard is a $\delta$-hydroxy nitrate (Rindelaub et al., 2016b), with expected relatively lower polarity and sensitivity for $I^-$ adduction compared to its $\beta$-analogue, as observed for the $\beta$- and $\delta$-hydroxy nitrates of isoprene (Iyer et al., 2016). We expect the $\beta$-ocimene hydroxy nitrates consist of both sensitive $\beta$-hydroxy nitrates and less sensitive configurations, complicating their quantitation. Thus, further study is warranted, for example quantum chemical calculations and laboratory experiments (Iyer et al., 2016; Lopez-Hilfiker et al., 2016), to examine how the degree of unsaturation and acyclic nature of the $\beta$-ocimene hydroxy nitrate isomers affect their binding affinities with $I^-$. $O_3$ measurements were not possible due to interference in the ozone monitor signal by $\beta$-ocimene (Walker and Hawkins, 1952). However, for the NO and $NO_2$ concentrations present in the chamber, and the $J_{NO_2}$ value ($7 \times 10^{-4}\,s^{-1}$), the calculated steady-state $O_3$ concentration was $< 1$ ppb (considering loss via reaction with NO and $\beta$-ocimene) and was not an important reactant in the experiments. Photochemistry was terminated by turning off the chamber lights when one-third of the NO concentration remained in the chamber to ensure that $\beta$-ocimene

oxidation occurred via OH oxidation only, and all $RO_2$ radicals reacted only with NO (Atkinson et al., 2006; Capouet et al., 2004; Rindelaub et al., 2015).

A series of blank and control experiments were conducted to evaluate (i) chamber reactions without $\beta$-ocimene or NO, (ii) gas- and particle-phase product loss to chamber walls under different RH conditions, and (iii) dark reactions of $NO_2$ with $\beta$-ocimene. Chamber reactions without $\beta$-ocimene or NO served as blank experiments and confirmed that all the $RONO_2$ detected during oxidation experiments were produced from the OH-initiated oxidation of $\beta$-ocimene in the presence of NO. Additionally, the particle-phase $RONO_2$ yields were corrected for wall loss from a second set of control experiments ($k_{wall\,SOA} = 4.3(\pm0.3) \times 10^{-5}\,s^{-1}$). The gas-phase wall loss rate constant for organic nitrates was determined based on observation of the first-order loss of the CIMS-determined monoterpene hydroxy nitrate ($M = C_{10}H_{17}NO_4$) signal in the dark ($[M + 1]^-$; $m/z = 342$; $k_{ONg} = 8.8(\pm2.2) \times 10^{-6}\,s^{-1}$). These experiments were conducted at varying relative humidities, and the wall loss rate constants ($k_{wall\,SOA}$, $k_{ONg}$) were determined from the loss in particle mass concentration and gas-phase concentration over time after the chamber lights were turned off. Blank experiments involving sampling from a cleaned chamber reveal no detectable degassing of organic nitrates from the walls, likely due to hydrolytic loss of adsorbed organic nitrates on the acidic walls (e.g., from uptake of $HNO_3$). All experimental data were corrected for dilution for both the photochemistry experiments, and for the post-experiment sampling, based on the sampling time, flow rate, and makeup gas flow rate utilized during each experiment ($k_{avg\,dilution} = 1.4(\pm0.1) \times 10^{-5}\,s^{-1}$ during the experiments and $4.6(\pm0.5) \times 10^{-5}\,s^{-1}$ during sampling). To further confirm that all $RONO_2$ were a product of the OH-initiated oxidation in the presence of NO, the third control experiment was conducted by injecting $NO_2$ into the dark chamber (UV lights remained off). The concentration of $NO_2$ and ocimene were monitored over time, and the gas- and particle-phase extracts were analyzed for the presence of organic nitrates.

The gas- and particle-phase products were separated and collected for offline analysis by sampling through a XAD4 resin-coated eight-channel annular denuder (URG-200) followed by a 47 mm PTFE particle filter. The denuder and filter were separately extracted in tetrachloroethylene ($\geq 99.9\,\%$, Sigma-Aldrich) immediately following a chamber experiment and stored in a refrigerator at $5\,^\circ C$. The extracts were then analyzed for total organic nitrate concentration using Fourier transform infrared spectroscopy (FTIR, Thermo Nicolet 6700) using a 1.00 cm liquid cell, as described by Rindelaub et al. (2015). A quality control experiment was conducted to determine if degradation of $RONO_2$ occurred when the samples were stored in the refrigerator, and no change in concentration was observed over a 24 h period. Extracts were analyzed by FTIR within 24 h of extraction and stored in a refrigerator when not in use to min-

imize any degradation or additional reactions in situ post-extraction. The organic nitrate concentrations were determined using the asymmetric -$ONO_2$ stretch at $1640\,cm^{-1}$ (Nielsen et al., 1995), using an absorption coefficient of $12\,900\,L\,mol^{-1}\,cm^{-1}$. An uncertainty of 7 % was applied to the FTIR measurements to account for the uncertainty in the absorption coefficient when using a proxy (ethyl hexyl nitrate in this study) for -$ONO_2$ quantitation (Carrington, 1960). After the FTIR analysis, samples were concentrated to near-complete dryness with ultra-high-purity nitrogen for use in high-resolution mass spectrometry analysis for structure elucidation and hydrolysis studies.

A series of quality control experiments were performed to evaluate the gas-phase organic nitrate collection efficiency by the denuder, particle transmission efficiency through the denuder, and particle collection efficiency on the filter, as described in Slade et al. (2017). To determine collection efficiency by the denuder, $\sim 200\,L$ PFA–Teflon bags of known concentrations of ethyl hexyl nitrate were prepared and the concentrations were measured using GC-FID before entering the denuder and immediately after. The collection efficiency of the denuder was determined to be $100(\pm 9)$ %. The filter collection efficiency ($96(\pm 19)$ %) was determined by measuring the particle mass concentration before and after the filter. Extraction efficiencies from both the denuder ($99(\pm 9)$ %) and filter ($100(\pm 2)$ %) were determined by measuring the concentration of $RONO_2$ in 50 mL aliquots of serial extracts, for actual experiments. To account for sample loss during concentration, different aliquots of the gas- and particle-phase extracts were analyzed at varying points during the solvent evaporation process. As a result, we correct for losses of the organic nitrates due to evaporation during solvent concentration, which contributes a 14 % uncertainty to the final concentration. That uncertainty is included in the overall concentration uncertainty discussed below.

To better understand the oxidation products, denuder and filter extracts were analyzed for their chemical composition via ultra-high-performance liquid chromatography with electrospray ionization time-of-flight tandem mass spectrometry (UPLC-ESI-ToF-MS/MS, Sciex 5600+ TripleToF with Shimadzu 30 series pumps and autosampler) (Slade et al., 2017). The separation was achieved using reverse-phased liquid chromatography (Phenomenex Kinetex EVO C18 column, 100 Å, $100\,mm \times 2.1\,mm$, 5 µm) following the gradient elution described in Surratt et al. (2008), and MS analyses were completed using negative and positive electrospray ionization (ESI) modes with multiple-reaction monitoring in the W reflectron geometry mode. The mass resolution ($m/\Delta m$) of this mass spectrometer is 25 000 at $m/z$ 195 using an accumulation time of 50 ms. The denuder and filter extracts were reconstituted in a 1 : 1 $v/v$ solvent mixture of HPLC-grade methanol with 0.1 % acetic acid and HPLC-grade water. The gradient elution was performed at $0.3\,mL\,min^{-1}$ with a binary mobile phase system: (A) 0.1 % acetic acid in water and (B) 0.1 % acetic acid in methanol. The 12 min gradient elu-

**Table 2.** Buffer systems used for hydrolysis studies and their corresponding pHs.

| Experimental pH | Buffer | pKa |
|---|---|---|
| $7.01 \pm 0.03$ | $H_2PO_4^- / HPO_4^{2-}$ | 7.2 |
| $4.10 \pm 0.01$ | $CH_3COOH / CH_3COO^-$ | 4.8 |
| $2.45 \pm 0.03$ | $HSO_4^- / SO_4^{2-}$ | 2.0 |

tion was as follows: the concentration of eluent B was 0 % for the first 2 min, linearly increased to 90 % from 2 to 10 min, held at 90 % from 10 to 10.2 min, and then decreased back to 0 % from 10.2 to 12 min (Surratt et al., 2008).

For hydrolysis studies, three experiments were selected representing dry, mid-RH, and high-RH conditions. The denuder and filter extracts were combined to form a bulk (gas- and particle-phase) $\beta$-ocimene nitrate solution that was then concentrated by solvent evaporation. The mixture was then reconstituted in 2.0 mL of an aqueous buffer solution at pH = 2.5, 4.1, and 7.0. To facilitate a direct comparison to results obtained in Rindelaub et al. (2015), the same buffer solutions were used, listed in Table 2. The solution was agitated using a magnetic stir bar, and 50 µL was extracted every 2 min. The extract was then mixed with 50 µL of methanol (99.9 %, Fisher Chemical). The resulting solution was stirred and immediately analyzed (to avoid any impact of continued hydrolysis) via direct-infusion ESI into a Thermo Scientific LTQ XL Linear Ion Trap mass spectrometer operated in negative ionization mode. The hydrolysis rate constants were determined from the first-order decay of various organic nitrate-related functional groups monitored by $MS^2$ fragmentation experiments. Briefly, fragmentation of the isolated molecular ion $m/z$ 214.1 ($C_{10}H_{16}NO_4$, expected first-generation oxidation product) was monitored at the instrument setting of collisional dissociation energy of 35 eV. The $m/z$ 46 ($NO_2^-$) and 62 ($NO_3^-$) mass fragments were used to quantify relative concentrations of $C_{10}H_{16}NO_4$ species as a function of the hydrolysis time. The exponential decrease in peak area of $m/z$ 46 ($NO_2^-$) was used to quantify the observed hydrolysis rate constant, which is effectively the weighted average for the different structural isomers present in the solution, weighted by their relative concentrations. The $m/z$ 62 ($NO_3^-$) fragment was used as a confirmation ion.

## 3 Results and discussion

### 3.1 Organic nitrate yields

A primary objective for this study is quantitative measurements of organic nitrate yields from OH reaction with $\beta$-ocimene, where the yields are defined as the concentration of $RONO_2$ produced ($\Delta RONO_2$ in ppb) relative to the concentration of BVOC consumed ($\Delta BVOC$ in ppb), i.e., $Y_{RONO_2} = \Delta RONO_2/\Delta BVOC$ (Slade et al., 2017), corrected for losses

of the organic nitrates. In our experiments, all peroxy radicals react with NO. Thus

$$RO_2 + NO \rightarrow RO + NO_2, \qquad (1a)$$

$$RO_2 + NO \rightarrow RONO_2. \qquad (1b)$$

$-d[\beta\text{-ocimene}]/dt = k_2[RO_2][NO]$, where $k_2 = k_{2a} + k_{2b}$, and $d[RONO_2]/dt = k_{2b}[RO_2][NO]$. Thus, $(d[RONO_2]/dt)/(-d[\beta\text{-ocimene}]/dt) = k_{2b}/k_2$. A plot of $\Delta[RONO_2]$ vs. $-\Delta[\beta\text{-ocimene}]$ yields the $RO_2$-weighted average "branching ratio", $k_{2b}/k_2$, for the $RO_2$ radicals produced when OH reacts with $\beta$-ocimene in the presence of $O_2$. We utilized the method of Kwok and Atkinson (1995) to estimate the fraction of time that OH adds to any of carbon no. 1, 2, 3, 4, 6, or 7, with numbers as shown here.

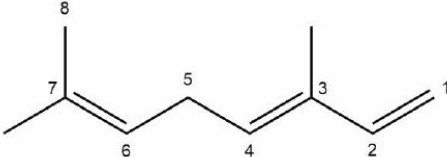

(E)-3,7-dimethylocta-1,3,6-triene

Because of the multiple double bonds at which OH can add, and because of the resonance structures of the multiple allylic radicals produced when OH adds to C no. 1 or 4 (as shown for the most prevalent addition point, C4, in Fig. 2), there are 10 possible main organic nitrate isomers, for each stereoisomer, i.e., the *cis*- or *trans*-$\beta$-ocimene, complicating the product analysis. Additional isomers may form from H abstraction from the C5 methylene group, but these are of less significance than the isomers produced from OH addition to the double bond. The measured total hydroxy nitrate, gas-, and particle-phase total organic nitrate yields under different RH conditions are shown in Fig. 3. Yields are corrected for dilution in the chamber, loss to the chamber walls, loss during preconcentration, and hydroxy nitrate consumption by reaction with OH (Rindelaub et al., 2015; Slade et al., 2017). The error bars on individual points reflect the propagated analytical uncertainties of the yields, incorporating all known uncertainties. The analytical uncertainty accounts for analytical errors associated with all the steps involved in sample collection, extraction, and analysis, including denuder and filter extraction and collection efficiencies, uncertainty in the FTIR calibration when using a proxy for -$ONO_2$ quantitation, dilution, wall loss, and consumption by OH. In Fig. 3, we see that the observed total $RONO_2$ yield and that for both gas and aerosol phases, and for the CIMS-determined hydroxy nitrates, are RH dependent, decreasing by $\sim \times 2$ over the range studied ($\sim 3$–70 % RH). The decrease indicates that hydroxy nitrates partition to the aerosol phase, where they are lost by hydrolysis, which was previously observed in the case of $\alpha$-pinene (Rindelaub et al., 2015). At higher RH, there will be more water associated with the particles, and the hydroxy nitrates are then more particle soluble (Rindelaub et al., 2015). If we then extrap-

olate to 0 % RH, the extent of particle-phase loss will be minimized, and thus the intercept represents the determined lower limit to the yield, assumed to represent the fraction of peroxy radicals reacting with NO that react to produce $RONO_2$. The intercept for the total $RONO_2$ yield, as shown in Fig. 3, is 38($\pm 9$) %, represented by the $y$ intercept of the black symbols in Fig. 3a. In Fig. 4, we present a summary of a range of previous observations of branching ratios for alkanes, alkenes, isoprene, and the two terpenes for which we have determined the total yield (assumed $RO_2$-weighted average branching ratio). It is shown that the branching ratio for $\beta$-ocimene- and $\alpha$-pinene-derived $RONO_2$ follows the trend observed for simple alkanes, within the uncertainty of the measurements (Arey et al., 2001; Rindelaub et al., 2015; Teng et al., 2015, 2017; Xiong et al., 2015).

We found that the first-generation hydroxy nitrates exist primarily in the gas phase (red markers in Fig. 3a), which is consistent with the low SOA yield, shown in Fig. 5. Although the total $RONO_2$ yield of $\beta$-ocimene is greater than that for $\alpha$-pinene oxidation, the particle-phase organic nitrate yield is much larger in the $\alpha$-pinene case (Rindelaub et al., 2015). This is likely due to the fact that the SOA yield is much larger for $\alpha$-pinene ($34 \pm 12$ %; Rindelaub et al., 2016a), since, as discussed in the next section, $\beta$-ocimene oxidation produces smaller, more volatile acyclic oxidation products. However, some of the low particle-phase $RONO_2$ yield must occur because of the hydrolysis in the aerosol phase. The relatively low SOA yield for $\beta$-ocimene can be attributed to the alkoxy radical decomposition products that produce the remaining 62 % of the first-generation oxidation products following the OH-initiated pathway in the presence of $NO_x$, for example as shown producing methyl vinyl ketone and 2-methyl-2-pentenal in Fig. 2. These decomposition products are expected to be primarily C6 and smaller carbonyl compounds, of much greater volatility than the $\beta$-ocimene hydroxy nitrate. This occurs in the case of this linear tri-alkene, because scission of the C–C bond at the $\alpha$ position to the alkoxy radical breaks the carbon chain into smaller chain carbonyl compound products, as shown in Fig. 2. As previously stated, the expected first-generation oxidation products (Fig. 2) are in the upper range of semi-volatile species, confirming their preferential presence in the gas phase (Donahue et al., 2011).

The CIMS-measured hydroxy nitrate yields (green markers in Fig. 3a) are lower than the total gas-phase $RONO_2$ yield (from FTIR), likely due to a lower CIMS sensitivity to the $\beta$-ocimene hydroxy nitrates than assumed, using the $\alpha$-pinene hydroxy nitrate standard as a proxy. However, it is important to note that we expect all organic nitrates in this system to be hydroxy nitrates. In this regard, it is important that the CIMS-observed hydroxy nitrate concentration decayed to the same relative extent as for the FTIR-determined total, as shown in Fig. 3. When the hydroxy nitrate undergoes hydrolysis, the reaction proceeds likely via SN1 unimolecular nucleophilic substitution, as discussed in Rindelaub et al. (2015). The nitrooxy functional group serves as

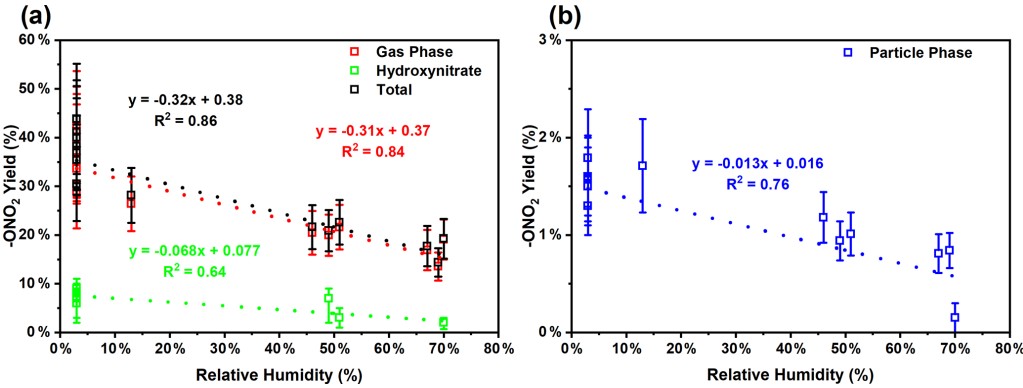

**Figure 2.** Example reaction pathways of the OH-initiated oxidation of $\beta$-ocimene in the presence of NO$_x$.

**Figure 3. (a)** Wall loss and dilution-corrected total gas-phase organic nitrate (-ONO2), total (both gas- and particle-phase) organic nitrate (quantified by FTIR), and hydroxy nitrate (quantified by CIMS) yield at different RH. **(b)** Wall loss and dilution-corrected total particle-phase organic nitrate yield at different RH. Error bars represent the propagated analytical uncertainty of the yields.

a leaving group and is replaced with a hydroxyl group, ultimately forming a diol. We calculated the vapor pressure of the diol produced from hydrolysis of compound A in Fig. 2, using SIMPOL (Pankow and Asher, 2008), and obtained $5.5 \times 10^{-8}$ atm at 20 °C, which is identical to that calculated for compound A itself. However, it is likely that the Henry's law constant for dissolution of the diol into water will be greater than that of compound A (Shepson et al., 1996), effectively increasing the partitioning into the aerosol phase.

## 3.2 SOA yields

Aerosol-mass-dependent secondary organic aerosol yields ($Y_{SOA}$) were calculated using the change in aerosol mass concentration ($\Delta M$ in µg m$^{-3}$) relative to the $\beta$-ocimene mass consumed ($\Delta$BVOC in µg m$^{-3}$), i.e., $Y_{SOA} = \Delta M / \Delta BVOC$ (Slade et al., 2017). The aerosol mass concentration was derived from the SMPS data, assuming spherical particles of $1.25$ g cm$^{-3}$ density (based on Ng et al., 2006), as shown in Fig. 1. The measured SOA yields as a function of aerosol mass concentration and chamber RH are shown in Fig. 5a, where the error bars reflect the propagated analytical uncertainty of the yields when considering the wall loss and dilution correction. While there is an apparent increase in aerosol yield with increasing humidity, it is not statistically significant (Fig. 5a). We calculate from the data in Fig. 5a that the SOA yield from OH oxidation of $\beta$-ocimene is less than

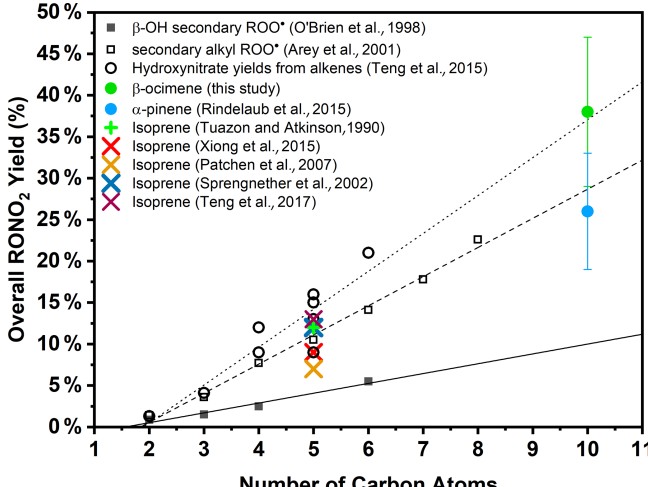

**Figure 4.** RONO$_2$ yield vs. size for alkyl peroxy radicals. Trend lines are representative for the data sets investigated in O'Brien et al. (1998) ($\beta$-hydroxy nitrates from $\beta$-hydroxyperoxy radicals), Arey et al. (2001) (alkyl nitrates from secondary alkyl peroxy radicals, alkanes), and Teng et al. (2015) (absolute hydroxy nitrate yields from alkenes).

1 % for typical aerosol mass concentrations in a moderately polluted forested environment, $\sim \leq 10\,\mu g\,m^{-3}$ (Hallquist et al., 2009). These yields are much lower than reported SOA yields from $\alpha$-pinene ($34 \pm 12\,\%$) (Rindelaub et al., 2015). Again, we hypothesize that this is related to structural differences of the alkoxy radicals, as shown in Figs. 2 and 6, specifically that, in the $\beta$-ocimene case, cleavage of the $\alpha$-C–C bond leads to two smaller radical fragments (Fig. 2) and ultimately smaller, more volatile carbonyl compounds (Gaona-Colmán et al., 2018; Reissell, 2002). For $\alpha$-pinene, most of the products retain a C10 backbone (Rindelaub et al., 2016a) due to ring-opening reaction resulting from the $\alpha$-carbon–carbon bond cleavage. To better understand the SOA yields, two-product absorptive partitioning model fits of the data from this study were compared to model fits and SOA yields of similar acyclic, tri-olefinic monoterpenes, shown in Fig. 5b (Böge et al., 2013; Griffin et al., 1999; Hoffmann et al., 1997). The two-product model fit parameters are as follows: $\alpha_1 = 0.1768$, $\alpha_2 = 0.0139$, $k_1 = 0.0020$, and $k_2 = 0.1698$. On average, the SOA yields in this study at all relative humidity values are less than that measured from $\beta$-ocimene photooxidation in a previous study, likely due to the absence of O$_3$ in our study (Hoffmann et al., 1997). Additionally, three experiments were conducted without seed aerosol to investigate the role of the inorganic seed aerosol. It was found that there is no apparent dependence of the yields in the seeded experiments versus the non-seeded experiments. We find that the SOA yields for myrcene (also a linear triene) were greater than those for $\beta$-ocimene (Böge et al., 2013). This is likely because of the two terminal double bonds on

myrcene that would result in a greater fraction of C9 carbonyl compound products.

Low-volatility, water-soluble organic nitrates can partition to the particle phase and contribute to the SOA mass. The estimated vapor pressure ($V_P$) of the $\beta$-ocimene hydroxy nitrate is $5.6 \times 10^{-8}$ atm at $20\,°C$ (Pankow and Asher, 2008), which is in the upper range of semi-volatile species, explaining the low aerosol-phase organic nitrate yields. Furthermore, the average saturation mass concentration ($C^*$) of $\beta$-ocimene hydroxy nitrates determined from the low-RH ($< 3\,\%$) experiments in this study was found to be $4000\,\mu g\,m^{-3}$, which is in agreement for species that exist almost entirely in the gas phase (Seinfeld and Pandis, 2016). This is in contrast to our experiments with $\alpha$-pinene-derived hydroxy nitrates, for which most were in the aerosol phase, at low relative humidities (Rindelaub et al., 2015). The estimated $V_P$ of the $\alpha$-pinene hydroxy nitrate shown in Fig. 6 is $8.6 \times 10^{-8}$ atm at $20\,°C$ (Pankow and Asher, 2008), which is greater than that of the $\beta$-ocimene hydroxy nitrate because of the ring in the C10 structure. However, the oxidation of $\alpha$-pinene results in ring opening rather than fragmentation (as shown in Fig. 6) in the case of ocimene. Thus, the first-generation oxidation products from $\alpha$-pinene will tend to be C10 species of lower vapor pressure than for the lower-carbon-number products in the ocimene case. The lower-vapor-pressure higher-carbon-number products likely lead to the greater aerosol yield ($\sim \times 6$) for $\alpha$-pinene, which in turn increase the partitioning of organic nitrates to the aerosol phase (Rindelaub et al., 2015). It is now well known that aerosol liquid water can also influence the SOA growth. In the ambient environment, as relative humidity increases (and as aerosol ages), the aerosol liquid water also increases (Carlton and Turpin, 2013), driving uptake of water-soluble oxidized organic compounds. However, in the photochemical smog chamber used in this study, the extent of VOC oxidation is very low (i.e., with mostly primary products, by design), and the resulting aerosols likely have relatively low O : C, resulting in a small correlation between relative humidity and aerosol yield, as shown in Fig. 5.

The gas- and particle-phase extracts were analyzed using UPLC-ToF-MS/MS to elucidate oxidation product structures. In Fig. 7, the selected ion chromatograms of expected oxidation products from ESI negative mode are presented for both the filter (Fig. 7a) and denuder (Fig. 7b) extracts for experiment no. 6. Tentative structures of the selected ions are shown, but the structures of these molecules were not confirmed with authentic standards. The arrangement of functional groups may vary depending on the position of the addition of the hydroxyl group to the original $\beta$-ocimene reactant. As shown in the light blue trace in Fig. 7, we do observe the dihydroxy dinitrate, although the effective gas-phase concentration could not be determined. As described above, we do correct for loss of the primary hydroxy nitrate by reaction with OH. In the case of the dinitrate, production would increase the effective measured organic nitrate yield. On the

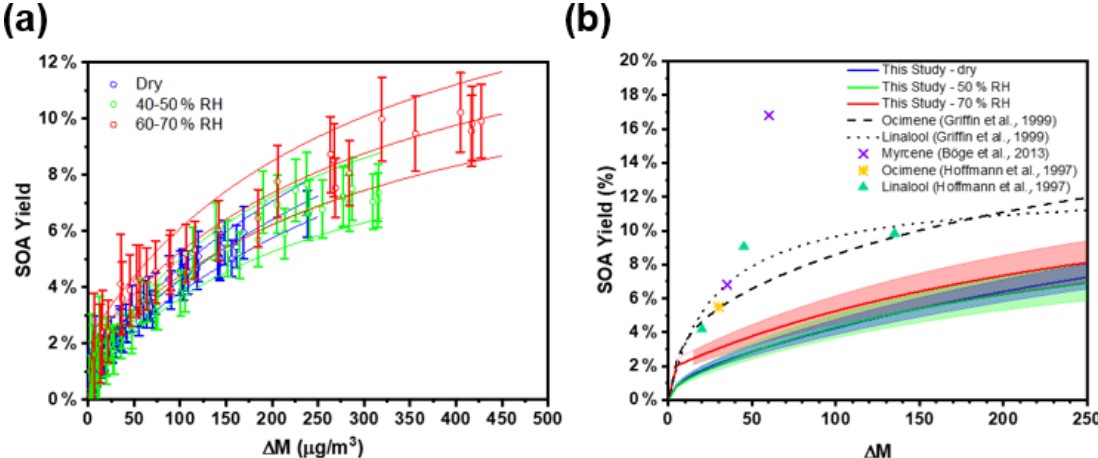

**Figure 5. (a)** Change in PM mass concentration ($\Delta M$) and secondary organic aerosol yields for seeded experiments under different RH conditions. The solid lines indicate the two-product absorptive partitioning model fit, and the dashed lines represent the 95 % confidence intervals of the fitting function. **(b)** Two-product absorptive partitioning model fits of this study vs. model fits and SOA yield percentage reported in the literature for the acyclic tri-olefinic and oxygenated terpenes. The shaded regions represent the 95 % confidence intervals of the fitting function. The fitting parameters are as follows: $\alpha_1 = 0.1768$, $\alpha_2 = 0.0139$, $k_1 = 0.0020$, and $k_2 = 0.1698$.

**Figure 6.** Example reaction pathways of the OH-initiated oxidation of $\alpha$-pinene in the presence of $NO_x$.

other hand, we also observe some of the trihydroxy nitrate, as shown in Fig. 7, indicating partial hydrolysis of the dinitrate. In this case, the product is neutral in its impact on the measured organic nitrate yield, and as discussed below, given the hydrolysis rate constants and the timescale of the experiments, it is likely that there is some hydrolysis of the primary organic nitrates to the corresponding diol, leading to the negative slope observed in Fig. 3. We hypothesize that the smaller (C6 or C4) fragments produced during the oxidation (not seen in our analysis) were lost during the solvent evaporation process, due to their high volatility.

### 3.3 Aqueous-phase hydrolysis

The observed decrease of the $RONO_2$ yields with increasing RH (Fig. 3) indicates acid-catalyzed hydrolysis in the aerosol phase. To date, there have only been hydrolysis rate constant

determinations for a few organic nitrates (Baker and Easty, 1950; Bean and Hildebrandt Ruiz, 2016; Boyd et al., 2015; Darer et al., 2011; Fisher et al., 2016; Hu et al., 2011; Liu et al., 2012; Pye et al., 2015; Zare et al., 2018), including the $\alpha$-pinene hydroxy nitrate (Rindelaub et al., 2015) and a range of isoprene-derived hydroxy nitrates (Jacobs et al., 2014). Here we studied the aqueous-phase hydrolysis kinetics (in bulk aqueous solution, adjusted for pH) of ocimene nitrates by examining the decay of the gas- and particle-phase organic nitrate products using mass spectrometry. In Table 3, we tabulated the hydrolysis rate constants reported in the literature for a wide range of organic nitrates, along with those determined here, indicating the solution pH, where known. There are a number of observations that can be made, most clearly that tertiary nitrates generally have very short hydrolysis lifetimes, that a $\beta$-hydroxy group shortens the hydrolysis life-

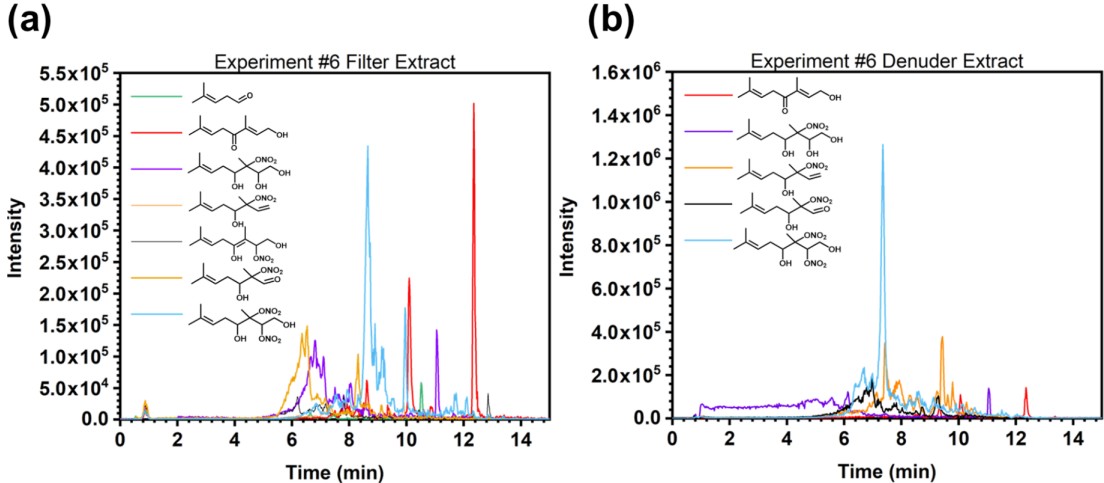

**Figure 7.** Selected ion chromatograms of expected oxidation products in electrospray (ESI) negative mode for experiment no. 6 in the **(a)** filter extract and **(b)** denuder extract. Structures listed are tentative assignments of neutral molecules based on the molecular formula of the selected ions. Actual structures may vary depending on the primary addition of the OH.

time, and that the hydrolysis lifetimes decrease with decreasing pH. We conducted measurements of the aqueous hydrolysis rate constants for the sum of the $\beta$-ocimene nitrates from the combined denuder and filter extracts for all experiments with RH $< 3\%$, via electrospray MS$^2$, as a function of the solution pH. In Fig. 8, we plot $\ln A_o / A_t$ vs. $t$, in seconds, where 0 is an arbitrary starting point for the pH-adjusted (buffered) samples. The resulting plots were linear, as expected for a first-order exponential decay. The slope of each line is equal to the first-order hydrolysis rate constant. The symbols in this figure represent replicates for each pH, and each panel is a different pH, all on the same scale for each pH. As shown in Fig. 8, the rate constants increase significantly, i.e., by $\sim \times 12$, over the range pH $= 7.0$ to pH $= 2.5$. These bulk rate constants represent those for the yield-average weighted organic nitrates produced. As there are many structural isomers produced during the OH-initiated oxidation of $\beta$-ocimene (Fig. 2), additional work is necessary exploring the structural dependence on hydrolysis rates. The initial concentration in the bulk solution is likely dominated by tertiary hydroxy nitrates, but the concentration and ionic strength for our experiments would be very low compared to those present within individual particles. This may impact the results, as discussed by Liu et al. (2020). The hydrolysis rates for primary and secondary hydroxy nitrates are likely slower than those of tertiary hydroxy nitrates, but increased ionic strength in the particle phase could contribute to more rapid hydrolysis and shorter lifetimes (Bean and Hildebrandt Ruiz, 2016; Boyd et al., 2015; Darer et al., 2011; Fisher et al., 2016; Hu et al., 2011; Jacobs et al., 2014; Liu et al., 2012, 2020; Pye et al., 2015; Takeuchi and Ng, 2019; Zare et al., 2018). Additional work investigating the hydrolysis rates of individual isomers (primary, secondary, tertiary) would be useful to understand the structural dependence of hydrolysis rates. It is, however,

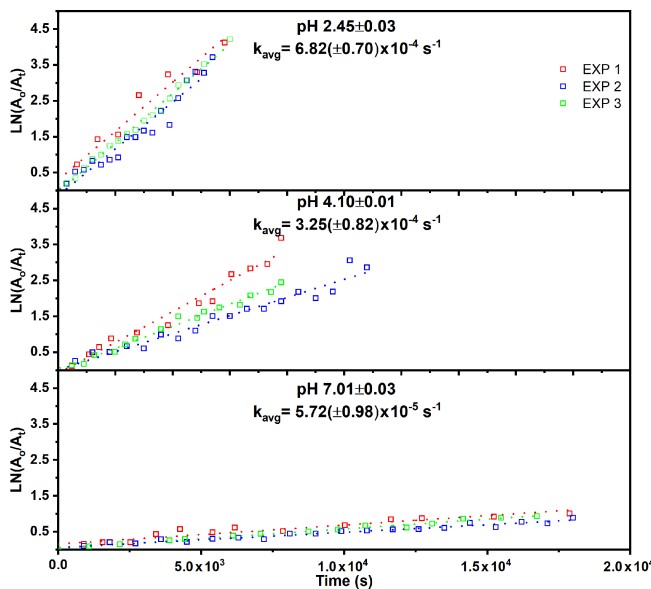

**Figure 8.** Hydrolysis rate constants for the bulk (gas- and particle-phase) organic nitrates at different pHs. Each pH was analyzed in triplicate, and the $k_{avg}$ is the average slope of each of the three trials.

interesting and useful that the linear chain nitrates studied here have very similar hydrolysis rates constants to those for the cyclic structure (with a distant -OH) of the $\alpha$-pinene hydroxy nitrate studied by Rindelaub et al. (2015).

In Fig. 9, we plot our data for the $\beta$-ocimene nitrates, along with those for the $\alpha$-pinene nitrate studied by Rindelaub et al. (2015), as a function of pH. The equation for the best-fit line in Fig. 9 is $\log_{10} k = -(0.24(\text{pH}) + 2.58)$, for $\beta$-ocimene. This shows that the natural lifetime against hydrolysis is 51 min at pH $= 4$ and decreases to 11 min at pH $= 1$.

**Table 3.** Hydrolysis rate constants and lifetimes reported in the literature.

| Species | Hydrolysis rate ($s^{-1}$) | $\tau_{hydrolysis}$ (h) | Type of study | Study |
|---|---|---|---|---|
| all tertiary and non-tertiary $RONO_2$[a] | $2.78 \times 10^{-4}\,s^{-1}$ | 1 | field study + model | Fisher et al. (2016) |
| tertiary $RONO_2$ from trimethyl benzene[a] | $4.63 \times 10^{-5}\,s^{-1}$ | 6 | chamber | Liu et al. (2012) |
| methyl nitrate[c] | $2.14 \times 10^{-8}\,s^{-1}$ | 13 000 | bulk | Baker and Easty (1952) |
| ethyl nitrate[c] | $1.07 \times 10^{-8}\,s^{-1}$ | 26 000 | | |
| ethyl nitrate[b] | $1.11 \times 10^{-4}\,s^{-1}$ | 2.5 | bulk | Hu et al. (2011) |
| isopropyl nitrate[c] | $5.24 \times 10^{-8}\,s^{-1}$ | 5300 | bulk | Baker and Easty (1952) |
| isopropyl nitrate[b] | $1.26 \times 10^{-4}\,s^{-1}$ | 2.2 | bulk | Hu et al. (2011) |
| isopropyl nitrate, pH 0.25 | $9.92 \times 10^{-6}\,s^{-1}$ | 28 | bulk | Rindelaub et al. (2016b) |
| isopropyl nitrate, pH 1.7 | $3.31 \times 10^{-7}\,s^{-1}$ | 840 | | |
| isopropyl nitrate, pH 4.0 | $3.86 \times 10^{-7}\,s^{-1}$ | 720 | | |
| isopropyl nitrate, pH 6.9 | $< 4.82 \times 10^{-8}\,s^{-1}$ | > 5760 | | |
| tert-butyl nitrate[c] | $3.19 \times 10^{-3}\,s^{-1}$ | 0.087 | bulk | Baker and Easty (1952) |
| isobutyl nitrate, pH 0.25 | $1.12 \times 10^{-5}\,s^{-1}$ | 23 | bulk | Rindelaub et al. (2016b) |
| isobutyl nitrate, pH 1.7 | $3.51 \times 10^{-7}\,s^{-1}$ | 792 | | |
| isobutyl nitrate, pH 4.0 | $4.13 \times 10^{-7}\,s^{-1}$ | 672 | | |
| isobutyl nitrate, pH 6.9 | $< 4.82 \times 10^{-8}\,s^{-1}$ | > 5760 | | |
| 1-nitrato-2,3-butanediol | $< 1.11 \times 10^{-7}\,s^{-1}$ | > 2500 | bulk | Darer et al. (2011) |
| 1-nitrooxy-2-hydroxy butane[b] | $1.63 \times 10^{-4}\,s^{-1}$ | 1.7 | bulk | Hu et al. (2011) |
| 2-nitrooxy-1-hydroxy butane[b] | $1.46 \times 10^{-4}\,s^{-1}$ | 1.9 | | |
| 2-nitrooxy-3-hydroxy butane[b] | $1.85 \times 10^{-4}\,s^{-1}$ | 1.5 | | |
| 2-nitrooxy-2-methyl-3-hydroxy propane[b] | $9.26 \times 10^{-3}\,s^{-1}$ | 0.03 | | |
| 3-nitrooxy-2-hydroxy-3-methyl butane[b] | $9.92 \times 10^{-3}\,s^{-1}$ | 0.028 | | |
| 2-nitrooxy-3-hydroxy-2,3-methyl butane[b] | $1.46 \times 10^{-2}\,s^{-1}$ | 0.019 | | |
| tertiary isoprene ON[c] | $9.26 \times 10^{-5}\,s^{-1}$ | 3 | field study + model | Zare et al., 2018 |
| 1-hydroxy-4-nitrooxy isoprene[c] | $6.76(\pm0.09) \times 10^{-3}\,s^{-1}$ | 0.04 | bulk | Jacobs et al. (2014) |
| 4-hydroxy-3-nitrooxy isoprene[c] | $1.59(\pm0.03) \times 10^{-5}\,s^{-1}$ | 17.5 | | |
| 1-hydroxy-2-nitrooxy-3-butene[c] | $9.95(\pm0.30) \times 10^{-6}\,s^{-1}$ | 27.9 | | |
| 2-methyl-2-nitro-1,3,4-butanetriol[c] | $4.15 \times 10^{-4}\,s^{-1}$ | 0.67 | bulk | Darer et al. (2011) |
| 3-methyl-1-nitro-2,3,4-butanetriol[c] | $< 1.11 \times 10^{-7}\,s^{-1}$ | > 2500 | | |
| 2-methyl-2-nitro-1,4-butanediol[c] | $4.55 \times 10^{-3}\,s^{-1}$ | 0.061 | | |
| 3-methyl-3-nitro-1,2-butanediol[c] | $2.31 \times 10^{-3}\,s^{-1}$ | 0.12 | | |
| $\beta$-pinene ON (from $NO_3\cdot$)[a] | $(6.17\text{–}9.26) \times 10^{-5}\,s^{-1}$ | 3-4.5 | chamber | Boyd et al. (2015) |
| tertiary $\beta$-pinene ON[c] | $9.26 \times 10^{-5}\,s^{-1}$ | 3 | field study + model | Pye et al. (2015) |
| $\alpha$-pinene ON[a] | $2.31 \times 10^{-5}\,s^{-1}$ | 12 | chamber | Bean and Hildebrandt Ruiz (2016) |
| $\alpha$-/$\beta$-pinene ON, pH 4.60 | $> 5.56 \times 10^{-4}\,s^{-1}$ | < 0.5 | chamber | Takeuchi and Ng (2019) |
| $\alpha$-pinene hydroxy nitrate, pH 0.25 | $1.98 \times 10^{-3}\,s^{-1}$ | 0.14 | bulk | Rindelaub et al. (2016b) |
| $\alpha$-pinene hydroxy nitrate, pH 1.0 | $3.81 \times 10^{-6}\,s^{-1}$ | 0.73 | | |
| $\alpha$-pinene hydroxy nitrate, pH 6.90 | $3.16 \times 10^{-5}\,s^{-1}$ | 8.80 | bulk | Rindelaub et al. (2015) |
| $\alpha$-pinene hydroxy nitrate, pH 4.00 | $2.14 \times 10^{-4}\,s^{-1}$ | 1.30 | | |
| $\alpha$-pinene hydroxy nitrate, pH 2.50 | $5.05 \times 10^{-4}\,s^{-1}$ | 0.55 | | |
| $\beta$-ocimene hydroxy nitrate, pH 7.01(±0.03) | $5.72(\pm0.41) \times 10^{-5}\,s^{-1}$ | 4.9 ±0.8 | bulk | This study |
| $\beta$-ocimene hydroxy nitrate, pH 4.13(±0.01) | $3.25(\pm0.12) \times 10^{-4}\,s^{-1}$ | 0.85 ±0.22 | bulk | |
| $\beta$-ocimene hydroxy nitrate, pH 2.45(±0.03) | $6.82(\pm0.36) \times 10^{-4}\,s^{-1}$ | 0.4 ±0.05 | bulk | |

[a] pH of the solution was not given. [b] For primary and secondary systems, the acid-catalyzed mechanism hydrolysis lifetime (for 55 wt % D2SO4 solution) is given. [c] Neutral hydrolysis lifetime is given.

These are very short compared to the lifetime of fine-mode aerosol, i.e., ∼ 1 week. Thus, we might expect the ambient aerosol-phase concentration of such hydroxy nitrates to be typically quite low, even though their uptake from the gas phase may have contributed significantly to aerosol mass, because they have subsequently transformed to diols. We also see that the hydrolysis lifetimes are very similar for the quite structurally different cyclic $\alpha$-pinene nitrate and acyclic $\beta$-ocimene nitrates, shown in Table 3. This implies that these values may be close to representative for a range of hydroxy nitrates. This rapid aqueous-phase hydrolysis points to the possible underestimation of aerosol-phase organic nitrates under ambient conditions (Ditto et al., 2020). As hydroxy nitrate production can represent an important fate for daytime oxidation of isoprene and terpenes in forested environments, the uptake of these compounds into acidic aerosol followed by hydrolysis can represent an important mechanism for conversion of $NO_x$ to $HNO_3$, as discussed in Romer Present et al. (2020) and Zare et al. (2018).

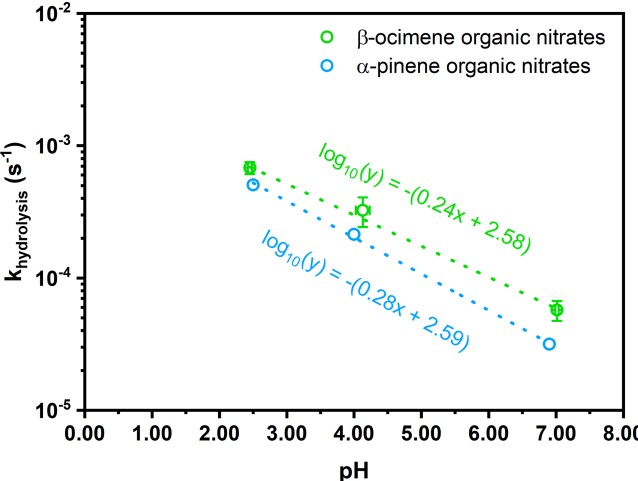

**Figure 9.** Hydrolysis rate constant for $\beta$-ocimene hydroxy nitrates (this study, green) and $\alpha$-pinene hydroxy nitrates (Rindelaub et al., 2015, blue) as a function of pH. The lines of best fit are listed in the colors corresponding to each monoterpene organic nitrates. Error bars represent the standard deviation between measurements of both the hydrolysis rate constant ($y$ error) and pH ($x$ error).

While the fate of the resulting diol in solution is unclear, there are a range of possible processes in the aqueous phase, including OH attack and oligomerization. Further oxidation of these olefinic diols would likely result in fragmentation, which might then cause product release into the gas phase (Otto et al., 2017). Diols, and these olefinic diols, can also undergo oligomerization to form even-lower-volatility products, which may have profound effects on the overall aerosol properties, such as viscosity and rate of diffusion (Glasius and Goldstein, 2016; Slade et al., 2019). The presence of oligomers from diols has previously been identified, but molecular characterization is quite limited (Stropoli et al., 2019). Careful investigation of the structures of these oligomers is necessary to develop reliable predictive models of SOA formation (Budisulistiorini et al., 2017; Pye et al., 2013; Surratt et al., 2007).

## 4 Atmospheric implications and conclusions

The production of organic nitrates serves as an important sink for gas-phase $NO_x$. The first-generation organic nitrate products are of relatively high vapor pressure and, for the polyunsaturated terpene $\beta$-ocimene, are subject to further gas-phase oxidative loss, to produce smaller but more oxidized products (Bateman et al., 2011). While the main products of that oxidation will be hydroxy nitrooxy carbonyl compounds of lower carbon number, and thus perhaps higher vapor pressure, the yield of dihydroxy dinitrates will be significant. The latter compounds will be of much lower vapor pressure; for example that for compound B shown in Fig. 2 is $2.4 \times 10^{-12}$ atm at 20 °C (Pankow and Asher, 2008), low

enough that it should rapidly undergo uptake into the aerosol phase, followed by hydrolysis to produce a 10-carbon tetrol and $NO_3^-$. However, the extent to which the gas-phase oxidation of the first-generation olefinic hydroxy nitrates re-releases $NO_x$ needs to be studied from laboratory experiments with the pure compounds. The organic nitrates will undergo hydrolysis in the particle phase, transforming from a hydroxy nitrate to a diol. The nitrooxy group will be released as the nitrate ion, which can impact regional nitrogen cycling by either remaining in the particle phase (at certain pHs) or being released as gas-phase nitric acid (at very high particle acidities). In either case, $NO_x$ has been permanently removed from the system (Galbavy et al., 2007; Rindelaub et al., 2015; Romer Present et al., 2020; Suarez-Bertoa et al., 2012; Zafiriou and True, 1979). Therefore, knowledge of the hydrolysis rate constants, especially for isoprene nitrates, is badly needed. The relatively large organic nitrate yield ($38 \pm 9$ %) in combination with the rapid aqueous-phase hydrolysis tends to make organic nitrate production an important sink for $NO_x$ in terpene-impacted forest environments. In addition, the light-dependent emission rate of $\beta$-ocimene suggests that in dense forest environments, like that of the University of Michigan Biological Station (Pratt et al., 2012), there could be steep vertical gradients in organic nitrate concentrations between canopy and ground, exceeding that of the other terpene nitrates modeled in Schulze et al. (2017). It is thus necessary to further assess the organic nitrate yield values, hydrolysis rate constants, and major oxidation products for a range of terpenes to fully understand the impact of their oxidation on the fate and distribution of $NO_x$.

These results suggest that $\beta$-ocimene hydroxy organic nitrates may be an important sink for gas-phase $NO_x$ in forest environments. Further work should be done to examine the structural dependence of hydroxy nitrate hydrolysis kinetics, for example the difference between $\beta$-hydroxy and $\delta$-hydroxy nitrates. Previous work has determined thermodynamic stabilities of structurally different organic nitrates (primary, secondary, tertiary) based on acid-catalyzed hydrolysis, where tertiary organic nitrates are less stable than primary or secondary organic nitrates at atmospherically relevant pHs, suggesting potential differences in hydrolysis rates between $\beta$-hydroxy and $\delta$-hydroxy nitrates (Hu et al., 2011). Differences between the hydrolysis rates of $\beta$-ocimene organic nitrates from $NO_3$ and OH-initiated oxidation should be explored, as significant differences in the hydrolysable fractions have been shown between organic nitrates formed from $NO_3$ and OH-initiated oxidation of both $\alpha$- and $\beta$-pinene (Takeuchi and Ng, 2019), and daytime $NO_3$ concentrations can be relevant in forests like that of University of Michigan Biological Station (Pratt et al., 2012; Schulze et al., 2017). It is necessary to understand the SOA components produced during this oxidation to better predict aqueous-phase processing. Further work on the condensed-phase chemistry of alcohols and olefinic oxygenated compounds in the aerosol phase is necessary.

*Data availability.* Experimental data are available upon request to the corresponding author.

*Author contributions.* JHS and PBS designed the research, and AL helped design the hydrolysis experiment approach. Experiments were conducted by ACM, TJ, and JHS, who did all the sample and data analysis. All authors contributed to the manuscript preparation and editing.

*Competing interests.* The authors declare that they have no conflict of interest.

*Acknowledgements.* We would like to thank Chloé de Perre and Linda Lee of the Department of Agronomy at Purdue University for the use of UPLC-MS instrumentation and analysis assistance. We also would like to thank Tad Kleindienst and John Offenberg of the EPA for assistance with the denuder-based filter setup. Anusha Hettiyadura and Krissy Morgan are thanked for assistance with hydrolysis studies and chamber experiments. We acknowledge Hartmut Hedderich of the Jonathan Amy Facility for Chemical Instrumentation (JAFCI, Purdue University) for access to FTIR instrumentation and Tim Miller (Birck Nanotechnology Center, Purdue University) for nano-pure water. This material is based upon work supported by the National Science Foundation Graduate Research Fellowship Program under grant no. DGE-1333468 and an NSF grant no. 1550398-CHE in support of the Shepson group. Any opinions, findings, and conclusions or recommendations expressed in this material are those of the author(s) and do not necessarily reflect the views of the National Science Foundation.

*Financial support.* This research has been supported by the National Science Foundation, Division of Atmospheric and Geospace Sciences (grant no. 1550398-CHE), and the National Science Foundation Graduate Research Fellowship Program (grant no. DGE-1333468).

*Review statement.* This paper was edited by Eleanor Browne and reviewed by three anonymous referees.

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
