# Peer review of "The Production and Hydrolysis of Organic Nitrates from OH Radical Oxidation of β-Ocimene"

_Atmospheric Chemistry and Physics, 2020_

## Referee Comment (RC1) · Anonymous Referee #1 · 11 Jul 2020

Summary:

This paper reports novel measurements of alkyl nitrate and SOA yield from OH + ocimene under high NOx conditions, and assesses hydrolysis rates, which are highly relevant to whether ambient SOA measurements accurately reflect the fraction of aerosol that partitioned to the particle phase via organic nitrates. FTIR functional group analysis enables a total organic nitrate yield measurement, while ESI-MS measurements also enable identification of potential individual nitrate product structures but are not quantitative and thus not possible to determine mechanism branching ratios. These new measurements contribute valuable data to a growing a literature on nitrate

and SOA yields and hydrolysis rates.

Major comments: 1) In the "Atmospheric Implications" section you mention some previous literature on acid catalyzed hydrolysis of nitrates, where tertiary nitrates were found to hydrolyse more rapidly than primary or secondary. This context is useful and I would suggest mentioning it earlier where you report the hydrolysis rate, and using your mechanism figure to call the reader's attention to this structural dependence of hydrolysis rates by pointing out the tertiary / secondary / primary ocimene nitrates.

2) On p. 5 near bottom: Do you expect similar sensitivity to ocimene hydroxynitrates as apin? What uncertainty?

3) Around line 142: refer to these as "blank and control experiments"? Not all "blanks" as you've described. Also, maybe give values for the gas and particle phase wall loss rates used to correct the data (or perhaps show a figure of uncorrected and corrected in a supplemental info, so readers can see how big the effect is.

4) P. 7: How did you "stop" hydrolysis after collecting the 50 ul aliquots every 2 minutes from your hydrolysis experiments?

5) Around line 314: Include (or add to SI and refer to) information on all of the uncertainties described here as going into the propagated error.

6) Around line 290, where you're discussing the lower vapor pressure a-pinene products: are the apin products expected to be lower Pvap simply because of the rings? Wouldn't both be C10 hydroxynitrates? Or are you thinking of the aggregated product mix that would include decomposition products?

7) You mention CIMS measurements of hydroxynitrates and report yields, but that time-series data is not in any figure. Would be nice to show other than the averages for each experiment in Fig. 3, if it's an online measurement. Could maybe add to Fig. 1?

8) Looking at Fig. 2, I would why wouldn't the non-conjugated double bond be the major radical attack, since it doesn't disrupt the conjugation of the other 2 double bonds. Also

in that figure, are the reactants over the top right arrow (OH, NO) correct? Needs a couple more steps maybe. And, the production of the alkoxy radical via rxn with NO in the middle of the second line should come from the peroxy radical (above it and to the right) and not the alkyl radical (directly above it). Finally, one of the structures is labeled "(A)" but no others are labeled. Wouldn't this be the same MW and indistinguishable in ESI-MS from the product on the third line? Some additional brief discussion of the detected structures and connection between this scheme and the chromatograms in Fig 7 would be welcome (maybe labelling more of the structures and putting the correspondence in Fig. 7 caption?)

9) In Figure 6 as in Fig. 2, I believe the NO rxn to produce alkoxy should come from the peroxy radical to the right and not the alkyl radical. But, I wonder if this figure is necessary in a paper focused on the ocimene reaction. Maybe for Fig. 6 you can just mention pinonaldehyde's MW, point out that there aren't obvious other fragmentation pathways, and reference Rindelaub?

Minor / technical suggestions and edits:

1) Line 34 : suggest moving number to read: "atmosphere (<1%) is significantly lower than reported yields..." to be clear that the % is the yield and not the amount lower than other yields.

2) Line 42: suggest change to "daytime monoterpenes not currently included..." to clarify that you are referring to this specific not included MT, and not stating that daytime MTs as a whole are not included in models.

3) Line 49: "which lead to the"

4) Line 57: "upon further oxidation and decomposition"

5) Line 59: "partitioning into the condensed phase"

6) Line 60: "Ambient measurements in forests have indicated"

7) Line 63: "lifetime of alkyl nitrates in the aerosol . . .. Aerosol itself to deposition (~1 week)"

8) Line 75: "Zare et al., 2018), which is sometimes rapid compared to measurements of plume aging." « and here is where you might add more about regiochemistry and hydrolysis rates, and what you expect for this alkene structure.

9) Line 94: replace "chemistry of these speces" with "hydrolysis rate" to be more specific?

10) Line 110: sounds like youre also using the evaporation setup for NO and HCHO, but I don't think they require evaporation – maybe reword?

11) Line 224: sub "yield" for "branching ratio"?

12) Lines 258-260: I find this a little confusing, the Henry's law constant of a diol increasing? Consider rewording.

13) Line 303: replace "such production" with "this reaction"? phrasing confused me.

14) Line 313: pH dependence, not RH dependence, right?

15) Line 315-316: list of previous hydrolysis rate constant expts is worded oddly, rework? And omit "Thus"

16) Line 330: suggest: "even though the nitrates have contributed. . . mass, because they have subsequently transformed to diols."

17) Line 332: looks to me like the similarities are not only at pH4 but across the range! Highlight this? When you call out the similarity at pH 4 it makes me think the agreement must have been worse at other PH's.

18) Line 346-347: suggest ". . . structure of these oligomers is necessary to develop reliable predictive. . ."

19) Line 351: suggest to remind the reader " and, for the polyunsaturated terpene

ocimene, are subject to. . ."

20) Line 356: "should rapidly undergo"

21) General note: the citation "Romer Present" is intermittently listed as 2019 and 2020, unify?

22) Fig 2 image quality is pretty poor, use a higher-res version?

23) Fig 4 : label of "Atmospheric pressure branching ratio" seems odd, maybe "Organic nitrate yield from OH oxidation of ocimene"?

24) Fig 5: "a) Relationship between PM mass. . .". Also, you determine the 2-product model fits but I don't think the parameters are reported anywhere in the paper, could add to this figure caption and make reference to in the text?
* * *

---

## Referee Comment (RC2) · Anonymous Referee #2 · 16 Jul 2020

Morales & Jayarathne et al. present laboratory study where B-Ocimene was oxidized by OH radicals in the presence of NO to form organic nitrates. The series of chamber experiments were used to investigate the hydroxynitrate yield and the impact of RH on the yield, SOA yield, and hydrolysis rates of the hydroxynitrates in the aerosol phase and the role of pH on hydrolysis. As there is minimal information about many of these processes for many compounds in the atmosphere, this study provides valuable constraints that will be beneficial for the atmospheric chemistry community. However, many technical aspects need to be described in better detail prior to publication. Upon addressing the comments here (and Reviewer #1, which has similar comments), the paper will be a valuable addition to ACP.

[Figure]

Major: Pg 4 Line 99 - 100: Why was the maximum RH only 70%, as this is below the deliquescent point of ammonium sulfate, the main seed used? How would you expect that to change the results discussed here, as this would inhibit aqueous chemistry? Further, why was ammonium sulfate the main seed used? Various studies have shown that forested environments have sulfate aerosol this is more like ammonium bisulfate (e.g., Weber et al., 2016, Nat. Geoscience), which may lead to different chemistry and uptake of the hydroxynitrates? Would this impact the aerosol yields and other aspects investigated during the chamber experiments? Finally, drawing more attention to the experiments that were allowed to self-nucleate (thus organic seed) maybe useful to discuss the role (or lack there of) or organic vs inorganic seed.

Pg 5, ln 133 - 135: It is stated that synthesized a-pinene hydroxynitrate standard was used to calibrate the CIMS for the ocimene hydroxynitrate. What is the uncertainty associated with this? Were any experiments conducted, comparing for example a voltage scan (e.g., Lopez-Hilfiker et al., 2016, AMT) or the standard and the species measured from the chamber, to compare the binding energies of the a-pinene and ocimene hydroxynitrate to determine if the sensitivity may be similar?

Pg 6, ln 144 - 147: Figures and an explanation of the amount of correction these experiments produced is needed. Is it in line with the general correction that has been observed in prior studies (e.g., Krechmer et al., EST, 2016; Krechmer et al., 2017, EST; Liu et al., 2019, AMT; Liu et al., 2019, Comms. Chem.; Matsunaga & Ziemann, 2010, AST)? Also, were the lines and instrument investigated for potential loss and corrected for (e.g., Deming et al., 2019, AMT; Pagonis et al., 2017, AMT)?

Pg 6, ln 150 - 152: During the quality control experiments to evaluate collection efficiency of the gas- and particle-phase species, were any experiments conducted or any observations made to determine if there were any losses or side reactions that occurred that would influence the results? How were the filters treated after collected the samples (immediately analyzed, stored for XX time in fridge, etc.) and would this impact the results?

Pg 7, ln 176 - 177: What acid/buffer solutions were used to make the bulk pH reactions? Also, were any experiments conducted with other type of acids, as the SN reaction may change? If not, can any comments be made about that?

In general, a subsection discussing all the uncertainty and how they are propagated into the error bars for the figures are extremely necessary. Right now, it is hard to discern how the error bars were calculated and what errors contributed/what the main error(s) were.

Pg 8, ln 212: Some of sources of uncertainty are mentioned here that were not brought up in the methods sections. A focused section that lays all the sources of uncertainty would help with these type of sentences.

Pg 12, ln 311 - 348: -Recent work has shown that the use of bulk solutions vs aerosol has large impacts on the chemistry, as aerosol is a much higher concentration solution, leading to higher ionic strengths and faster or slower reactions (Liu et al., 2020, PNAS). Wondering if this was considered for the results discussed here? If not, at least mention this limitation to raise awareness that the rates maybe different. -It was briefly mentioned, but further discussion about primary vs tertiary hydrolysis is warranted here. For example, since different type of nitrates are formed, do the results discussed here and shown in Fig. 8 indicate a predominance of tertiary nitrates and the generally rapid hydrolysis? Would a "second" line from primary expected at all, or are the yields for primary expected to be low enough to not have a noticeable 2nd, slower hydrolysis rate? -Finally, it is not clear which experiments in Table 1 were used for this section. Please specify.

Pg 13, ln 333 - 335: This sentence seems very misleading, as there have been numerous studies that has measured appreciable concentrations of organic nitrates, with different techniques, in various locations (Ayres et la., 2015; Fry et al., 2013; Fry et al., 2018; Lee et al., 2016; Kiendler-Scharr et al., 2016; Ng et al., 2017). I strongly recommend rephrasing.

Figure 7: It is not clear how the structures were determined.

Minor

Throughout the text, please check references. There are many locations where the Name et al., year or year do not have the proper parenthesis. Also, some of the references at do not have proper abbreviations for journals.

Pg 7, ln 172: was the solution linearly increased during the 2 to 10 min?

Pg 9, ln 226 - 231 & Fig 4: Were the results compared against the parameterization for yields from Carter and Atkinson (J. Atmos. Chem., 1989)? If not, it maybe valuable to do that comparison for reference for organic nitrates that have not been measured.

Pg 10, ln 257 (and others): It would be useful to also state what the C* is along with the vapor pressure, as many aerosol chemists think in C* and it provides an easy comparison about partitioning (e.g., C* 0 or 1 means a lot of the products will be in the aerosol-phase and are more semi-volatile but C* 4 - 6 mean that it's more intermediate-volatile and less likely to be in the aerosol-phase).

Fig 4: It's unclear what the different lines represent.

Fig 6: I agree with Reviewer #1 that this figure currently doesn't seem warranted, and it is currently mentioned in the text (once I think).

Other fig: Finally, I agree with Reviewer #1 that a figure showing the time series of the gas-phase species measured by the CIMS would also be useful (along with other parameters such as aerosol mass concentration, b-Ocimene, etc.).

———————————————

---

## Referee Comment (RC3) · Anonymous Referee #3 · 21 Jul 2020

Review of '**The Production and Hydrolysis of Organic Nitrates from OH Radical Oxidation of β-Ocimene**', by A. C. Morales and T. Jayarathne et al.

**Overview:**

This paper presents a detailed and well described laboratory study of the OH radical-initiated oxidation of β-ocimene, an atmospherically significant species, in the presence of NO. This study investigates the formation of organic nitrates, the impact of β-ocimene on secondary organic aerosol (SOA) yields, and the subsequent hydrolysis rate constants for the organic nitrate products in solutions of differing pH. The experimental methodology is well thought through and clearly presented.

The authors build upon a previous study of $\alpha$-pinene (Rindelaub et al., 2015), experimental work presented by the same research group as here and compare the results of this study to their previous study on the $\alpha$-pinene system.

Overall the manuscript is very well written, with experimental work described to a good level of detail. The script can be slightly repetitive at times, but the scientific message is clearly conveyed to the reader. The significance of this detailed work could be further highlighted by the authors, by emphasising the implications of the described chemistry in terms of human health.

**General Comments:**

The authors should take care in maintaining consistency in the text, for example always using "OH" over "•OH", and "$\alpha$-pinene" over "alpha-pinene", "β-ocimene" rather than just "ocimene". More detailed comments on text inconsistencies are given in the detailed comments section below.

The methods section would benefit from a little more discussion on how product loss to the chamber walls was accounted for. Would you expect any chemical reactions to occur at the wall? How do you account for losses to the chamber wall / dilution? Do you use an inert tracer species to examine loss effects?

Though their previous study on nitrate formation from $\alpha$-pinene + OH (Rindelaub et al., 2015) provides a useful comparison, sometimes the authors rely too heavily on referencing this study, and could expand more on their findings within this paper, without asking the reader to refer to their previous work.

The figure captions could provide more detail.

**Detailed Comments:**

Abstract, lines 30-31: Inconsistent spacing after 'pH', "pH=4, and 24($\pm$3) min at pH = 2.5"

Introduction, line 48: What is a 'criteria air pollutant'?

Introduction, line 48: This is a little unclear as BVOCs don't actually react with NOx. Maybe something like, "BVOCs participate in chemical reactions with the atmospheric oxidants OH, $NO_3$ and $O_3$, which leads…"

Introduction, line 54: affects

Introduction, line 54: delete full-stop after 'quality'

Introduction, line 71: change "ozone" to "$O_3$" for consistency with previous text

Methods, line 99: What is the uncertainty on the approximate experimental temperature quoted?

Methods, line 110: Replace "5 lmp" with "5 Lmin$^{-1}$"

Methods, line 146: How did you account for the product loss to the chamber walls? And how did you account for species dilution from the chamber? Did you use an inert tracer?

Results and Discussion, 3.1, several lines: Replace "Fig" with "Figure" for consistency

Results and Discussion, 3.1, line 220: Can this sentence be re-written as… In this study we did this… (Rindelaub et al., 2015).

Results and Discussion, 3.1, line 226: '…summary of a range of previous observations…'

Results and Discussion, 3.1, line 249: "…yield, likely…"

Results and Discussion, 3.1, line 258: Replace "compound A" with "Compound A" for consistency

Results and Discussion, 3.2, lines 271: Is there a reference for this value from a 'moderately pollluted forested environment'?

Results and Discussion, 3.2, lines 274 and 277: Replace "α-carbon-carbon" with "α-C-C" to maintain consistency with line 244

Results and Discussion, 3.2, lines 276: No need for 'e.g.' here

Results and Discussion, 3.2, lines 280/281: '…less but similar…', this doesn't convey useful information, can this be quantified?

Results and Discussion, 3.2, lines 289: No need for comma after 'humidities'

Results and Discussion, 3.3, lines 327 and 332: Replace "pH=4" with "pH = 4" for consistency

Results and Discussion, 3.3, line 328: Replace "pH=1" with "pH = 1" for consistency

Results and Discussion, 3.3, line 336: What does 'forest-impact environments' mean?

Atmospheric Implications and Conclusions, line 360: Remove "from"

Atmospheric Implications and Conclusions, line 361: Replace "as the nitrate ion" with "as a nitrate ion"

Atmospheric Implications and Conclusions, line 369: Define "UMBS"

Atmospheric Implications and Conclusions, line 386: "These results suggest that ocimene hydroxy organic nitrates may be an important sink for gas phase NOx in forest environments." seems out of place in the 'future work' paragraph of the conclusions. Perhaps start the paragraph with this before going into further detail on how future work could expand on your findings.

Acknowledgement, line 393: "We acknowledge Dr. Hartmut Hedderich of the (Jonathan Amy Facility for Chemical Instrumentation, Purdue University)" missing abbreviation, or remove brackets.

**Figures and Tables:**

Figure 2: The alkoxy radical in row 2 comes from reaction of the peroxy radical in row 1 with NO (not reaction of the alkyl radical with NO as currently shown)

Figure 2: It is not clear what reaction pathways the two arrows pointing from **(A)** are supposed to represent. Is the top pathway supposed to be:

I)      OH addition at C1 (as for the bottom pathway), followed by NO reaction with the peroxy radical formed forming the alkoxy radical which then forms a carbonyl?

or

II)      OH addition at C2, followed by NO reaction with the peroxy radical formed and scission of the C1-C2 bond?

Either way I think that the product formed should have one less carbon (i.e. the C1 carbon is lost and the carbonyl is on the C2 carbon).

Figure 4: Rename x axis from "#C" to "Carbon number", for clarity

Figure 6: As for Figure 2, the alkoxy radical should come from RO2+NO, not R + NO

Figure 9: In figure caption, replace "ocimene" with "β-ocimene" and "alpha-pinene" with "α-pinene" for consistency

---

## Author Comment (AC1) · 13 Oct 2020

**Response to reviewer comments: Reviewer #1**

The authors thank the reviewer for their helpful comments. The changes made to the manuscript and/or our replies to the reviewer are addressed below. The reviewer comments are shown in regular font, and the responses are in bold.

Major comments: 1) In the "Atmospheric Implications" section you mention some previous literature on acid catalyzed hydrolysis of nitrates, where tertiary nitrates were found to hydrolyse more rapidly than primary or secondary. This context is useful and I would suggest mentioning it earlier where you report the hydrolysis rate, and using your mechanism figure to call the reader's attention to this structural dependence of hydrolysis rates by pointing out the tertiary / secondary / primary ocimene nitrates.

**We thank the reviewer for calling attention to this. The structural hydrolysis is briefly mentioned in the introduction (lines 73-76) but is further discussed in section 3.3 Aqueous Phase Hydrolysis. Table 3 has been added to facilitate a direct comparison of all known/measured/assumed (modeled) literature values for a range of different organic nitrates. We feel that this is a helpful addition to the paper.**

**More specifically, additional discussion has been added on page 15 lines 391-404:**

***"As there are many structural isomers produced during the OH-initiated oxidation of β-ocimene (Figure 2), additional work is necessary exploring the structural dependence on hydrolysis rates. The initial concentration in the bulk solution is likely dominated by tertiary hydroxy nitrates, but the concentration and ionic strength for our experiments would be very low compared to those present within individual particles; this may impact the results, as discussed by Liu et al., 2020. The hydrolysis rates of primary and secondary hydroxy nitrates are likely slower than those of tertiary hydroxy nitrates, but increased ionic strengths in the particle phase could contribute to more rapid hydrolysis and shorter lifetimes (Bean and Hildebrandt Ruiz, 2016; Boyd et al., 2015; Darer et al., 2011; Fisher et al., 2016; Hu et al., 2011; Jacobs et al., 2014; Liu et al., 2012, 2020; Pye et al., 2015; Takeuchi and Ng, 2019; Zare et al., 2018). Additional work investigating the hydrolysis rates of individual isomers (primary, secondary, tertiary) would be useful to understand the structural dependence of hydrolysis rates. It is however interesting and useful that the linear chain nitrates studied here have very similar hydrolysis rates constants compared to those for the cyclic structure (with a distant -OH) of the α-pinene hydroxy nitrate studied by Rindelaub et al., 2015."***

2) On p. 5 near bottom: Do you expect similar sensitivity to ocimene hydroxynitrates as apin? What uncertainty?

**Thank you for this question. There is not enough data in the literature to provide a basis for an expectation. However, since we expect that all organic nitrates produced are hydroxy nitrates (though not all alpha, beta) given the ~x4 difference between the "hydroxynitrate" and total gas-phase organic nitrates from the FTIR data, it appears the instrument sensitivity we used (based on the Slade et al. (2017) α-pinene hydroxy nitrate sensitivity) was x4 too large. As we state in the text, we think the data (in Fig. 3) are useful in that the FTIR total**

**and the CIMS total decrease with humidity by the same extent, as we would expect if the FTIR and CIMS were measuring the same hydroxy nitrate species. This is discussed on page 10 lines 260-264 of the revision.**

3) Around line 142: refer to these as "blank and control experiments"? Not all "blanks" as you've described. Also, maybe give values for the gas and particle phase wall loss rates used to correct the data (or perhaps show a figure of uncorrected and corrected in a supplemental info, so readers can see how big the effect is.

**The sentence has been adjusted to say *"blank and control experiments"* on Page 6, Line 158.**

**The wall loss rate constant ($k_{wall\ SOA}$ = 4.3($\pm$0.3)$\times$10$^{-5}$ s$^{-1}$, line 163) and average dilution rate ($k_{avg.\ dilution}$ = 1.4($\pm$0.1)$\times$10$^{-3}$ min$^{-1}$, line 167) were included.**

4) P. 7: How did you "stop" hydrolysis after collecting the 50 ul aliquots every 2 minutes from your hydrolysis experiments?

**Aliquots were mixed with methanol and immediately analyzed to avoid the impact of continued hydrolysis. A sentence to this effect has been added on Page 9, line 222-225:**

*"The extract was then mixed with 50 µL of methanol (99.9%, Fisher Chemical). The resulting solution was stirred and immediately analyzed (to avoid any impact of continued hydrolysis) via direct infusion electrospray ionization (ESI) into a Thermo Scientific LTQ XL Linear Ion Trap mass spectrometer operated in negative ionization mode."*

5) Around line 314: Include (or add to SI and refer to) information on all of the uncertainties described here as going into the propagated error.

**Thank you for this very helpful comment. Additional paragraphs discussing all uncertainties and corrections were added to section 2 Methods, including discussion of propagation of the overall measurement uncertainty, on page 10, lines 255-261 of the revision.**

**More detail can be found regarding the blank and control experiments on Page 6 lines 158-171:**

*"A series of blank and control experiments was conducted to evaluate i) chamber reactions without β-ocimene or NO, ii) gas- and particle-phase product loss to chamber walls under different RH, and iii) dark reactions of $NO_2$ with β-ocimene. Chamber reactions without β-ocimene or NO served as blank experiments and confirmed that all the $RONO_2$ detected during oxidation experiments were produced from the •OH-initiated oxidation in the presence of NO. Additionally, the SOA yields were corrected for wall loss from the second set of control experiments ($k_{wall\ SOA}$ = 4.3($\pm$0.3)$\times$10$^{-5}$ s$^{-1}$). These experiments were conducted at varying relative humidities and the wall loss rate constant ($k_{wall\ SOA}$) was determined from the loss in particle mass concentration over time after the chamber lights were turned off. All experiments were corrected for dilution based on the sampling time, flow rate, and makeup gas flow rate utilized during each experiment ($k_{avg.\ dilution}$ = 1.4($\pm$0.1)$\times$10$^{-3}$ min$^{-1}$). To further confirm all RONO2 was a product of the •OH-initiated oxidation in the presence of NO, the third control experiment was conducted by injecting $NO_2$ into the dark chamber (UV lights remained off). The concentration*

*of NO₂ and ocimene were monitored over time, and the gas- and particle-extracts were analyzed for the presence of organic nitrates."*

**Discussion regarding the FTIR correction when using a proxy for calibration was added to Page 7 lines 183-185:**

*"An uncertainty of 7% was applied to the FTIR measurements to account for the uncertainty in the absorbtion coefficient when using a proxy (ethyl hexyl nitrate in this study) for -ONO2 quantitation (Carrington, 1960). After the FTIR analysis, samples were concentrated to near-complete dryness with ultra-high-purity nitrogen for use in high resolution mass spectrometry analysis for structure elucidation and hydrolysis studies."*

**Discussion regarding the collection and extraction efficiencies and their respective uncertainties was added to Page 7 lines 188-201:**

*"A series of quality control experiments was performed to evaluate the gas-phase organic nitrate collection efficiency by the denuder, particle transmission efficiency through the denuder, and particle collection efficiency on the filter, as described in Slade et al., 2017. To determine collection efficiency by the denuder, ~200-liter PFA-Teflon bags of known concentrations of ethyl hexyl nitrate were prepared and the concentrations were measured using GC-FID before entering the denuder and immediately after. The collection efficiency of the denuder was determined to be 100(±9)%. The filter collection efficiency (96(±19)%) was determined by measuring the particle mass concentration before and after the filter. Extraction efficiencies from both the denuder (99(±9)%) and filter (100(±2)%) were determined by measuring the concentration of RONO₂ in 50 mL aliquots of serial extracts, for actual experiments. To account for sample loss during concentration, different aliquots of the gas- and particle-phase extracts were analyzed at varying points during the solvent evaporation process. As a result, we correct for losses of the organic nitrates due to evaporation during solvent concentration, which contributes a 14% uncertainty to the final concentration. That uncertainty is included in the overall concentration uncertainty discussed below."*

6) Around line 290, where you're discussing the lower vapor pressure a-pinene products: are the apin products expected to be lower Pvap simply because of the rings? Wouldn't both be C10 hydroxynitrates? Or are you thinking of the aggregated product mix that would include decomposition products?

**It is not because the products have ring structures per se, but rather because when a ring structure alkoxy radical decomposes by scission of the alpha C-C, this opens the ring, and you still have 10 carbon atoms. For ocimene, when alkoxy radicals decompose this produces two smaller carbonyl compound products, as discussed in detail on page 12 and in Figures 2 and 6.**

**It is, however, also the case that the ring structure hydroxy nitrates have greater vapor pressures. The following sentence including the vapor pressure of a-pinene hydroxy nitrates was added to Page 13 lines 340-342:**

*"The estimated V$_P$ of the α-pinene hydroxy nitrate shown in Figure 6 is 8.6×10$^{-8}$ atm at 20°C (Pankow and Asher, 2008), which is greater than that of the β-ocimene hydroxy nitrate because of the ring in the C10 structure."*

7) You mention CIMS measurements of hydroxynitrates and report yields, but that timeseries data is not in any figure. Would be nice to show other than the averages for each experiment in Fig. 3, if it's an online measurement. Could maybe add to Fig. 1?

**Thank you, Figure 1 has been updated to include the CIMS measurement data.**

8) Looking at Fig. 2, I would why wouldn't the non-conjugated double bond be the major radical attack, since it doesn't disrupt the conjugation of the other 2 double bonds. Also in that figure, are the reactants over the top right arrow (OH, NO) correct? Needs a couple more steps maybe. And, the production of the alkoxy radical via rxn with NO in the middle of the second line should come from the peroxy radical (above it and to the right) and not the alkyl radical (directly above it). Finally, one of the structures is labeled "(A)" but no others are labeled. Wouldn't this be the same MW and indistinguishable in ESI-MS from the product on the third line? Some additional brief discussion of the detected structures and connection between this scheme and the chromatograms in Fig 7 would be welcome (maybe labelling more of the structures and putting the correspondence in Fig. 7 caption?)

**We believe that attack at carbon #4 is preferred in part from the greater stability of the resultant allylic radical. Figure 2 has been updated to correct the issues in the mechanism and the labels for structures (A) and (B) (discussed in the text, on pages 12 lines 300-304 and Page 16 lines 435-437) have been moved to be more visually obvious. Because the mass spectra are very similar for the isomeric organic nitrates, and we do not have standards for individual ocimene nitrates, we cannot unambiguously relate the ocimene hydroxy nitrates in the chromatograms in Figure 7 to specific structures.**

9) In Figure 6 as in Fig. 2, I believe the NO rxn to produce alkoxy should come from the peroxy radical to the right and not the alkyl radical. But, I wonder if this figure is necessary in a paper focused on the ocimene reaction. Maybe for Fig. 6 you can just mention pinonaldehyde's MW, point out that there aren't obvious other fragmentation pathways, and reference Rindelaub?

**Figure 6 has been updated to have the alkoxy radical as a product of the peroxy radical + NO. The mechanism shown in Figure 6 is included because of the discussion on Page 13 lines 341, and the ease of reading. Additional discussion regarding Figure 6 has been added to Page 13 lines 340-347:**

*"The estimated V$_P$ of the α-pinene hydroxy nitrate shown in Figure 6 is 8.6×10$^{-8}$ atm at 20°C (Pankow and Asher, 2008), which is greater than that of the β-ocimene hydroxy nitrate because of the ring in the C10 structure. However, the oxidation of α-pinene results in ring opening rather than fragmentation (as shown in Figure 6), in the case of ocimene. Thus, the first-generation oxidation products from α-pinene will tend to be C10 species of lower vapor pressure than for the lower carbon number products in the ocimene case. The lower vapor pressure higher carbon-number products likely lead to the greater aerosol yield (~×6) for α-pinene, which*

*in turn increased the partitioning of organic nitrates to the aerosol phase (Rindelaub et al., 2015)."*

Minor/technical suggestions and edits:

**All grammatical and syntax suggestions have been incorporated into the revised manuscript. Thank you. The remaining minor comments are addressed below:**

1) Line 34 : suggest moving number to read: "atmosphere (<1%) is significantly lower

than reported yields: : :" to be clear that the % is the yield and not the amount lower

than other yields.

**This change has been made to line 33.**

2) Line 42: suggest change to "daytime monoterpenes not currently included: : :" to

clarify that you are referring to this specific not included MT, and not stating that daytime

MTs as a whole are not included in models.

**Thank you for the suggestion. This change has been made to line 41.**

8) Line 75: "Zare et al., 2018), which is sometimes rapid compared to measurements

of plume aging." « and here is where you might add more about regiochemistry and

hydrolysis rates, and what you expect for this alkene structure.

**Additional discussion of the structural dependence of organic nitrate hydrolysis was added to Page 15 lines 390-404:**

*"These bulk rate constants represent those for the yield-average weighted organic nitrates produced. As there are many structural isomers produced during the OH-initiated oxidation of β-ocimene (Figure 2), additional work is necessary exploring the structural dependence on hydrolysis rates. The initial concentration in the bulk solution is likely dominated by tertiary hydroxy nitrates, but the concentration and ionic strength for our experiments would be very low compared to those present within individual particles; this may impact the results, as discussed by Liu et al., 2020. The hydrolysis rates of primary and secondary hydroxy nitrates are likely slower than those of tertiary hydroxy nitrates, but increased ionic strengths in the particle phase could contribute to more rapid hydrolysis and shorter lifetimes (Bean and Hildebrandt Ruiz, 2016; Boyd et al., 2015; Darer et al., 2011; Fisher et al., 2016; Hu et al., 2011; Jacobs et al., 2014; Liu et al., 2012, 2020; Pye et al., 2015; Takeuchi and Ng, 2019; Zare et al., 2018). Additional work investigating the hydrolysis rates of individual isomers (primary, secondary, tertiary) would be useful to understand the structural dependence of hydrolysis rates. It is however interesting and useful that the linear chain nitrates studied here have very similar hydrolysis rates constants compared to those for the cyclic structure (with a distant -OH) of the α-pinene hydroxy nitrate studied by Rindelaub et al., 2015."*

10) Line 110: sounds like youre also using the evaporation setup for NO and HCHO,

but I don't think they require evaporation – maybe reword?

**Thank you for this observation. The formaldehyde is 37% in water and requires evaporation, but the NO does not. The sentence, on Page 5 lines 115-118 was reworded to say:**

*"Formaldehyde (37% v/v in water, Sigma) was injected using the same set-up, followed by pure NO (99.5%, Praxair) injected using the same set-up without heat. Formaldehyde photolysis serves as an •OH precursor in the presence of NO (Possanzini et al., 2002)."*

15) Line 315-316: list of previous hydrolysis rate constant expts is worded oddly, rework?

And omit "Thus"

**We have decided to add Table 3, which lists those hydrolysis rate constants and lifetimes (as a function of pH) we could find in the literature. This is discussed on page 14 of the revision.**

17) Line 332: looks to me like the similarities are not only at pH4 but across the range!

Highlight this? When you call out the similarity at pH 4 it makes me think the agreement

must have been worse at other PH's.

**Table 3 has been added to facilitate a comparison across the entire pH range studied.**

21) General note: the citation "Romer Present" is intermittently listed as 2019 and

2020, unify?

**Thank you for this observation. This typo has been corrected.**

22) Fig 2 image quality is pretty poor, use a higher-res version?

**Figure 2 has been improved.**

23) Fig 4 : label of "Atmospheric pressure branching ratio" seems odd, maybe "Organic

nitrate yield from OH oxidation of ocimene"?

**The Figure 4 caption and Y axis have been changed to *"$RONO_2$ yield".***

24) Fig 5: "a) Relationship between PM mass: : :". Also, you determine the 2-product

model fits but I don't think the parameters are reported anywhere in the paper, could

add to this figure caption and make reference to in the text?

**The parameters have been included on page 12 lines 324-325 and in the figure caption for Figure 5.**

**Response to reviewer comments: Reviewer #2**

The authors thank the reviewer for their comments that improve the quality of the paper. The changes made to the manuscript and/or our repliers to the reviewer are addressed below. The reviewer comments are shown in regular font, the responses are in bold font.

Major: Pg 4 Line 99 - 100: Why was the maximum RH only 70%, as this is below the deliquescent point of ammonium sulfate, the main seed used? How would you expect that to change the results discussed here, as this would inhibit aqueous chemistry? Further, why was ammonium sulfate the main seed used? Various studies have shown that forested environments have sulfate aerosol this is more like ammonium bisulfate (e.g., Weber et al., 2016, Nat. Geoscience), which may lead to different chemistry and uptake of the hydroxynitrates? Would this impact the aerosol yields and other aspects investigated during the chamber experiments? Finally, drawing more attention to the experiments that were allowed to self-nucleate (thus organic seed) maybe useful to discuss the role (or lack there of) or organic vs inorganic seed.

**The maximum RH was only 70% because we found it difficult to humidify the chamber above that level. It now states on page 4 lines 104-105 of the revised manuscript that 70% was the maximum humidity achievable.**

**Ammonium sulfate was the main seed used to facilitate a direct comparison between Rindelaub et al., (2015) and this study. This has been clarified on Page 5 line 120.**

**During these experiments, there is very significant aerosol growth, some of which is organic, including organic acids, and come of which is nitric acid. In this process, we expect considerable acidification of the particles, and conversion of sulfate to bisulfate, so that there will be a mixture of the two in the highly complex particle phase (but which at the surface will likely be a purely organic mixture). We have added a comment on page 13 lines 327-330 of the revision, that there is no apparent dependence of the yields in the seed/no seed experiments, although the data are relatively sparse.**

*"Additionally, three experiments were conducted without seed aerosol to investigate the role of the inorganic seed aerosol. It was found that there is no apparent dependence of the yields in the seeded experiments versus the non-seeded experiments"*

Pg 5, ln 133 - 135: It is stated that synthesized a-pinene hydroxynitrate standard was used to calibrate the CIMS for the ocimene hydroxynitrate. What is the uncertainty associated with this? Were any experiments conducted, comparing for example a voltage scan (e.g., Lopez-Hilfiker et al., 2016, AMT) or the standard and the species measured from the chamber, to compare the binding energies of the a-pinene and ocimene hydroxynitrate to determine if the sensitivity may be similar?

**Thank you for this question. There is not enough data in the literature to provide a basis for an uncertainty estimate. However, since we expect that all organic nitrates produced are hydroxy nitrates (though not all alpha, beta) given the ~x4 difference between the "hydroxynitrate" and total gas-phase organic nitrates from the FTIR data, it appears the instrument sensitivity we used (based on the Slade et al. (2017) α-pinene hydroxy nitrate**

**sensitivity) was x4 too large.  We did not do the experiments that you suggested.  As discussed for reviewer #1, and as we state in the text, we think the data are useful in that the FTIR total and the CIMS total decrease with humidity by the same extent.  This is discussed on page 10 lines 260-264 of the revision.**

**We edited the relevant section on Page 6 lines 138-150 as follows:**

*"A synthesized α-pinene hydroxy nitrate standard was used as a surrogate standard to calibrate the CIMS for β-ocimene hydroxy nitrate determination (Rindelaub et al., 2016b; Slade et al., 2017). We note there may be differences in the sensitivity of the instrument towards individual β-ocimene hydroxy nitrate isomers compared to the α-pinene hydroxy nitrate standard due to differences in polarity and thus binding affinity with I- (Iyer et al., 2016; Lopez-Hilfiker et al., 2016). The synthetic α-pinene hydroxy nitrate standard is a δ-hydroxy nitrate (Rindelaub et al., 2016), with expected relatively lower polarity and sensitivity for I- adduction compared to its β-analogue, as observed for the β- and δ-hydroxy nitrates of isoprene (Iyer et al., 2016). We expect the β-ocimene hydroxy nitrates consist of both sensitive β-hydroxy nitrates and less sensitive configurations, complicating their quantitation. Thus, further study is warranted, e.g., quantum chemical calculations and laboratory experiments (Iyer et al., 2016; Lopez-Hilfiker et al., 2016) to examine how the degree of unsaturation and acyclic nature of the β-ocimene hydroxy nitrate isomers affect their binding affinities with I-."*

Pg 6, ln 144 - 147: Figures and an explanation of the amount of correction these experiments produced is needed. Is it in line with the general correction that has been observed in prior studies (e.g., Krechmer et al., EST, 2016; Krechmer et al., 2017, EST; Liu et al., 2019, AMT; Liu et al., 2019, Comms. Chem.; Matsunaga & Ziemann, 2010, AST)? Also, were the lines and instrument investigated for potential loss and corrected for (e.g., Deming et al., 2019, AMT; Pagonis et al., 2017, AMT)?

**Additional paragraphs discussing all uncertainties and corrections were added to section 2 Methods, pages 6-7 lines 158-201.**

**More detail can be found regarding the blank and control experiments on Page 6 lines 158-171 of the revision, as described for our response to Reviewer #1:**

**Discussion regarding the FTIR correction when using a proxy for calibration was added to Page 7 lines 183-185, as described above for Reviewer #1:**

**Discussion regarding the collection and extraction efficiencies and their respective uncertainties was added to Page 7 lines 188-201, as described above for Reviewer #1:**

Pg 6, ln 150 - 152: During the quality control experiments to evaluate collection efficiency of the gas- and particle-phase species, were any experiments conducted or any observations made to determine if there were any losses or side reactions that occurred that would influence the results? How were the filters treated after collected the samples (immediately analyzed, stored for XX time in fridge, etc.) and would this impact the results?

**The extracts were stored in a refrigerator and analyzed within 24 hours to minimize any losses or side reactions. Discussion can be found on Page 7 lines 174-181 of the revision:**

*"The denuder and filter were separately extracted in tetrachloroethlyene (≥99.9%, Sigma-Aldrich) immediately following a chamber experiment and stored in a refrigerator at 5°C. The extracts were then analyzed for total organic nitrate concentration using Fourier transform infrared spectroscopy (FTIR, Thermo Nicolet 6700) using a 1.00 cm liquid cell, as described by Rindelaub et al., 2015. A quality control experiment was conducted to determine if degradation of RONO2 occurred when the samples were stored in the refrigerator, and no change in concentration was observed over a 24-hour period. Extracts were analyzed by FTIR within 24 hours of extraction and stored in a refrigerator when not in use to minimize any degradation or additional reactions in situ post-extraction."*

Pg 7, ln 176 - 177: What acid/buffer solutions were used to make the bulk pH reactions? Also, were any experiments conducted with other type of acids, as the SN reaction may change? If not, can any comments be made about that? In general, a subsection discussing all the uncertainty and how they are propagated into the error bars for the figures are extremely necessary. Right now, it is hard to discern how the error bars were calculated and what errors contributed/what the main error(s) were.

**Table 2 has been added presenting the buffer solutions and their pKa values. We used the same buffer systems as Rindelaub et al., (2015) to facilitate a direct comparison between monoterpene hydroxy nitrate hydrolysis rate constants.**

**Error bars in Figure 9 are the standard deviations between each measurement in the three datasets. The Figure 9 caption has been revised to describe the error.**

Pg 8, ln 212: Some of sources of uncertainty are mentioned here that were not brought up in the methods sections. A focused section that lays all the sources of uncertainty would help with these type of sentences.

**Additional paragraphs discussing all uncertainties and corrections were added to the section 2 Methods.**

**More detail can be found regarding the blank and control experiments on Page 6 lines 158-171, as described above for Reviewer #1.**

**Discussion regarding the FTIR correction when using a proxy for calibration was added to Page 7 lines 183-185, as discussed above for Reviewer #1.**

**Discussion regarding the collection and extraction efficiencies and their respective uncertainties was added to Page 7 lines 188-201, as discussed above for Reviewer #1.**

Pg 12, ln 311 - 348: -Recent work has shown that the use of bulk solutions vs aerosol has large impacts on the chemistry, as aerosol is a much higher concentration solution, leading to higher ionic strengths and faster or slower reactions (Liu et al., 2020, PNAS). Wondering if this was considered for the results discussed here? If not, at least mention this limitation to raise awareness that the rates maybe different. -It was briefly mentioned, but further discussion about primary vs

tertiary hydrolysis is warranted here. For example, since different type of nitrates are formed, do the results discussed here and shown in Fig. 8 indicate a predominance of tertiary nitrates and the generally rapid hydrolysis? Would a "second" line from primary expected at all, or are the yields for primary expected to be low enough to not have a noticeable 2nd, slower hydrolysis rate? - Finally, it is not clear which experiments in Table 1 were used for this section. Please specify.

**Thank you for this insightful suggestion. We describe on page 15 of the revision that the experiments were done in bulk dilute solution (and in line 390 of the revision) and that that could impact the results, and cite Liu et al., as you suggest. We did not observe any time-dependent different trend for hydrolysis. We do expect the primary nitrates to be a small fraction of the total.**

**The rate constants were determined using the combined denuder and filter extracts for all experiments conducted at RH <3%, as described on Page 15 lines 382-385 of the revision.**

**More discussion of the hydrolysis rate constants was added to Page 15 lines 390-404:**

*"These bulk rate constants represent those for the yield-average weighted organic nitrates produced. As there are many structural isomers produced during the OH-initiated oxidation of β-ocimene (Figure 2), additional work is necessary exploring the structural dependence on hydrolysis rates. The initial concentration in the bulk solution is likely dominated by tertiary hydroxy nitrates, but the concentration and ionic strength for our experiments would be very low compared to those present within individual particles; this may impact the results, as discussed by Liu et al., 2020. The hydrolysis rates of primary and secondary hydroxy nitrates are likely slower than those of tertiary hydroxy nitrates, but increased concentrations and ionic strengths in individual particles could contribute to more rapid hydrolysis and shorter lifetimes (Bean and Hildebrandt Ruiz, 2016; Boyd et al., 2015; Darer et al., 2011; Fisher et al., 2016; Hu et al., 2011; Jacobs et al., 2014; Liu et al., 2012, 2020; Pye et al., 2015; Takeuchi and Ng, 2019; Zare et al., 2018). Additional work investigating the hydrolysis rates of individual isomers (primary, secondary, tertiary) would be useful to understand the structural dependence of hydrolysis rates. It is however interesting and useful that the linear chain nitrates studied here have very similar hydrolysis rates constants compared to those for the cyclic structure (with a distant -OH) of the α-pinene hydroxy nitrate studied by Rindelaub et al., 2015."*

**We note again, as for our response to Reviewer #1, we have put our results in the context of all available information in the literature by adding Table #3. We hope that you find that useful.**

Pg 13, ln 333 - 335: This sentence seems very misleading, as there have been numerous studies that has measured appreciable concentrations of organic nitrates, with different techniques, in various locations (Ayres et la., 2015; Fry et al., 2013; Fry et al., 2018; Lee et al., 2016; Kiendler-Scharr et al., 2016; Ng et al., 2017). I strongly recommend rephrasing.

**Thank you for this observation. The sentence has been revised to say underestimation instead of imply difficulty with ambient measurements on Page 16 lines 414-415:**

*"This rapid aqueous-phase hydrolysis can explain the possible underestimation of substantial aerosol phase organic nitrates under ambient conditions (Ditto et al., 2020)."*

Figure 7: It is not clear how the structures were determined.

**Thank you for this comment. The structures are tentatively assigned based on the molecular formula and expected oxidation products using the mechanism in Figure 2. A clarification has been added to the Figure 7 caption.**

Minor

Throughout the text, please check references. There are many locations where the Name et al., year or year do not have the proper parenthesis. Also, some of the references at do not have proper abbreviations for journals.

**We have checked this carefully.**

Pg 7, ln 172: was the solution linearly increased during the 2 to 10 min?

**Yes, this change has been made to Page 8 lines 214 of the revision.**

Pg 9, ln 226 - 231 & Fig 4: Were the results compared against the parameterization for yields from Carter and Atkinson (J. Atmos. Chem., 1989)? If not, it maybe valuable to do that comparison for reference for organic nitrates that have not been measured.

**We prefer to not do that for this figure, to not make the figure more complex, and because Arey et al. recommended that the Carter and Atkinson fitting parameters be updated given the data in their paper, which are shown in Figure 4. Additionally, the estimation method enables calculation as a function of temperature and pressure, which were not variable in our experiments.**

Pg 10, ln 257 (and others): It would be useful to also state what the $C^*$ is along with the vapor pressure, as many aerosol chemists think in $C^*$ and it provides an easy comparison about partitioning (e.g., $C^*$ 0 or 1 means a lot of the products will be in the aerosol-phase and are more semi-volatile but $C^*$ 4 - 6 mean that it's more intermediatevolatile and less likely to be in the aerosol-phase).

**Thank you for this insightful suggestion. The $C^*$ was calculated and discussed on Page 13 lines 336-339 of the revision:**

*"Furthermore, the average saturation mass concentration ($C^*$) of β-ocimene hydroxy nitrates determined from the low RH (<3%) experiments in this study was found to be 4000 μg m$^{-3}$, which is in agreement for species that exist almost entirely in the gas-phase (Seinfeld and Pandis, 2016)."*

Fig 4: It's unclear what the different lines represent.

**The Figure 4 caption has been updated to clarify.**

Fig 6: I agree with Reviewer #1 that this figure currently doesn't seem warranted, and it is currently mentioned in the text (once I think).

**Thank you for this observation. It was unintentional to omit discussion of Figure 6. Discussion of this figure has been included in the text on Page 13 lines 340-347:**

*"The estimated $V_P$ of the α-pinene hydroxy nitrate shown in Figure 6 is 8.6×10$^{-8}$ atm at 20˚C (Pankow and Asher, 2008), which is greater than that of the β-ocimene hydroxy nitrate because of the ring in the C10 structure. However, the oxidation of α-pinene results in ring opening rather than fragmentation (as shown in Figure 6), in the case of ocimene. Thus, the first-generation oxidation products from α-pinene will tend to be C10 species of lower vapor pressure than for the lower carbon number products in the ocimene case. The lower vapor pressure higher carbon-number products likely lead to the greater aerosol yield (~×6) for α-pinene, which in turn increased the partitioning of organic nitrates to the aerosol phase (Rindelaub et al., 2015)."*

Other fig: Finally, I agree with Reviewer #1 that a figure showing the time series of the gas-phase species measured by the CIMS would also be useful (along with other parameters such as aerosol mass concentration, b-Ocimene, etc.).

**Thank you, you are right. Figure 1 has been updated to include the CIMS measurement data.**

**Response to reviewer comments: Reviewer #3**

The authors thank the reviewer for their comments that improve the quality of the paper. The changes made to the manuscript and/or our repliers to the reviewer are addressed below. The reviewer comments are shown in regular fonts, the responses are in bold fonts.

General Comments:

**All grammatical and syntax suggestions have been made to the manuscript. The remaining comments are addressed below:**

The authors should take care in maintaining consistency in the text, for example always using "OH" over "•OH", and "α-pinene" over "alpha-pinene", "β-ocimene" rather than just "ocimene".

**This issue was fixed throughout the entire manuscript.**

More detailed comments on text inconsistencies are given in the detailed comments section below.

The methods section would benefit from a little more discussion on how product loss to the chamber walls was accounted for. Would you expect any chemical reactions to occur at the wall? How do you account for losses to the chamber wall / dilution? Do you use an inert tracer species to examine loss effects?

**Additional paragraphs discussing all uncertainties and corrections were added to section 2 Methods.**

**More detail can be found regarding the blank and control experiments on Page 6 lines 158-201 of the revision, as discussed for Reviewer #1 above.**

**Discussion regarding the FTIR correction when using a proxy for calibration was added to Page 7 lines 183-185 in the revision, as discussed above for Reviewer #1.**

**Discussion regarding the collection and extraction efficiencies and their respective uncertainties was added to Page 7 lines 188-201 in the revision, as discussed above for Reviewer #1.**

Though their previous study on nitrate formation from α-pinene + OH (Rindelaub et al., 2015) provides a useful comparison, sometimes the authors rely too heavily on referencing this study, and could expand more on their findings within this paper, without asking the reader to refer to their previous work.

**Thank you for this comment. We tried to address all instances that relied too heavily on the previous literature from this research group (Slade et al., 2017; Rindelaub et al., 2015, 2016a, 2016b).**

The figure captions could provide more detail.

**Thank you for this note. We added more detail to the figures 4, 5, 7 and 9 captions.**

Detailed Comments:

Abstract, lines 30-31: Inconsistent spacing after 'pH', "pH=4, and 24(±3) min at pH = 2.5"

**This has been fixed in line 30 of the revision.**

Introduction, line 48: What is a 'criteria air pollutant'?

**We have changed this to say** *"regulated air pollutants"* **in line 47.**

Introduction, line 48: This is a little unclear as BVOCs don't actually react with NOx. Maybe something like, "BVOCs participate in chemical reactions with the atmospheric oxidants OH, NO3 and O3, which leads…"

**This sentence has been clarified to state that BVOCs can react with NO₂, a component of NOx. The reference Tuazon and Atkinson 1990 (re NO₂ + isoprene) has been added. The new sentence can be found on Page 2 lines 47-52:**

*"BVOCs participate in chemical reactions with regulated air pollutants, including NO₂ and O₃, and with radical species such as OH and NO₃, which lead to the formation of low volatility oxygenated compounds that partition into aerosol particles and represent a source of secondary organic aerosol (SOA) (Atkinson and Arey, 2003; Hallquist et al., 2009; Hatakeyama et al., 1991; Isaksen et al., 2009; Lee et al., 2016; Monks et al., 2009; Perring et al., 2013; Pye et al., 2015; Tuazon and Atkinson, 1990)."*

Introduction, line 54: delete full-stop after 'quality'

**Thank you for this suggestion. The sentence, on Page 2 lines 52-53, now reads:**

*"Globally, the oxidation of BVOCs emitted from forests represents the largest source of SOA and affects climate and air quality (Hallquist et al., 2009)."*

Introduction, line 71: change "ozone" to "O3" for consistency with previous text
**This has been updated throughout the manuscript for consistency.**

Methods, line 99: What is the uncertainty on the approximate experimental temperature quoted?
**The uncertainty has been added on Page 4 lines 101-104:**
*"These experiments were conducted as a function of chamber RH, keeping other variables constant as possible, and at 24(±2) ºC."*

Methods, line 146: How did you account for the product loss to the chamber walls? And how did you account for species dilution from the chamber? Did you use an inert tracer?
**We caclculated loss of the hydroxy nitrates to the walls using the α-pinene nitrate as a proxy, as discussed in Ridelaub et al. (2015). We reference that work on lines 255-257, which state:**
*"Yields are corrected for dilution in the chamber, loss to the chamber walls, loss during preconcentration, and hydroxynitrate consumption by reaction with OH (Rindelaub et al., 2015; Slade et al., 2017)."* **Particle wall loss was calculated by conducting a chamber experiment and monitoring the decrease in particle mass concentration after the experiment has concluded. Dilution was calculated based on the experiment duration and the backup gas flow rate. More detail can be found regarding the blank and control experiments on Page 6 lines 158-171 of the revision, as described above for Reviewer #1.**

Results and Discussion, 3.2, lines 271: Is there a reference for this value from a 'moderately polluted forested environment'?

**The citation Hallquist et al., (2009) has been added on Page 12 line 315. We also modified that to read ~≤10 μg m$^{-3}$.**

Results and Discussion, 3.2, lines 280/281: '…less but similar…', this doesn't convey useful information, can this be quantified?

**This sentence has been adjusted on Page 12 lines 325-327 to read:**
***"On average, the SOA yields in this study at all relative humidity values are less than that measured from β-ocimene photooxidation in a previous study, likely due to the absence of O$_3$ in our study (Hoffmann et al., 1997)."***

Results and Discussion, 3.3, line 336: What does 'forest-impact environments' mean?

**This sentence has been adjusted on Page 16 line 416-417 to read:**
***"…isoprene and terpenes in forested environments…"***

Atmospheric Implications and Conclusions, line 369: Define "UMBS"

**This has been corrected on Page 17 lines 449:**
***"…like that of University of Michigan Biological Station (Pratt et al., 2012; Schulze et al., 2017)."***

Atmospheric Implications and Conclusions, line 386: "These results suggest that ocimene hydroxy organic nitrates may be an important sink for gas phase NOx in forest environments." seems out of place in the 'future work' paragraph of the conclusions. Perhaps start the paragraph with this before going into further detail on how future work could expand on your findings.

**Thank you for this suggestion. This line has been moved to the beginning of the paragraph on Page 17 lines 454-455 of the revision.**

Acknowledgement, line 393: "We acknowledge Dr. Hartmut Hedderich of the (Jonathan Amy Facility for Chemical Instrumentation, Purdue University)" missing abbreviation, or remove brackets.

**This section has been updated on Page 18 lines 473-474:**
***"We acknowledge Dr. Hartmut Hedderich of the Jonathan Amy Facility for Chemical Instrumentation (JAFCI, Purdue University)…"***

**Figures and Tables:**

Figure 2: The alkoxy radical in row 2 comes from reaction of the peroxy radical in row 1 with NO (not reaction of the alkyl radical with NO as currently shown)

**Thank you for this observation, the figure has been corrected.**

Figure 2: It is not clear what reaction pathways the two arrows pointing from (**A**) are supposed to represent. Is the top pathway supposed to be: I) OH addition at C1 (as for the bottom pathway), followed by NO reaction with the peroxy radical formed forming the alkoxy radical which then forms a carbonyl? Or II) OH addition at C2, followed by NO reaction with the peroxy radical

formed and scission of the C1-C2 bond? Either way I think that the product formed should have one less carbon (i.e. the C1 carbon is lost and the carbonyl is on the C2 carbon).
**Thank you for this observation. The mechanism has been corrected and Figure 2 has been updated.**

Figure 4: Rename x axis from "#C" to "Carbon number", for clarity
**Thank you, the figure has been updated.**

Figure 6: As for Figure 2, the alkoxy radical should come from RO2+NO, not R + NO
**Thank you for this observation, the figure has been corrected.**

Figure 9: In figure caption, replace "ocimene" with "β-ocimene" and "alpha-pinene" with "α-pinene" for consistency
**The figure caption has been updated.**

---

## Author Response (AR2)

Response to the editor:

Thank you for your suggestions. We have done our best to address every comment and have made all appropriate changes to the manuscript. The author's responses are in bold font.

Major comment:

Thank you for your careful consideration of the referee comments. I believe that the manuscript is improved and that nearly all of the referee comments have been sufficiently addressed. However, I think that the consideration of wall-loss of the gas-phase species needs further elaboration. Currently, the wall-loss for the gas-phase species is discussed via reference to a previous publication (Rindelaub et al). Given the community conversation around this topic currently, I think a brief discussion is warranted here as well.

**Thank you for this suggestion. We have carefully reviewed the detailed analysis of the corrections for wall loss, along with all other corrections, including those for dilution. While we had corrected for dilution during sampling this had not been done for the time frame of the photochemistry experiment. A complete reanalysis of all results with corrections has led to a revised total organic nitrate yield (and slightly larger uncertainty) of 38±9%. Page 6 line 162-174 of the revision has been edited to include a more detailed discussion of the experiments leading to our understanding of the appropriate gas-phase wall loss rate constant, and our experiments showing no detectable desorption of organic nitrates from previous experiments, as indicated below.**

***"Additionally, the particle-phase $RONO_2$ yields were corrected for wall loss from a second set of control experiments ($k_{wall\ SOA} = 4.3(\pm0.3)\times10^{-5}\ s^{-1}$). The gas-phase wall loss rate constant for organic nitrates was determined based on observation of the first-order loss of the CIMS-determined monoterpene hydroxynitrate (M = $C_{10}H_{17}NO_4$) signal in the dark ($[M + 1\ ]^-$; m/z = 342; $k_{ONg} = 8.8(\pm2.2)\times10^{-6}\ s^{-1}$). These experiments were conducted at varying relative humidities and the wall loss rate constants ($k_{wall\ SOA}$, $k_{ONg}$) were determined from the loss in particle mass concentration and gas-phase concentration over time after the chamber lights were turned off. Blank experiments involving sampling from a cleaned chamber reveal no detectable degassing of organic nitrates from the walls, likely due to hydrolytic loss of adsorbed organic nitrates on the acidic walls (e.g. from uptake of $HNO_3$). All experimental data were corrected for dilution for both the photochemistry experiments, and for the post-experiment sampling, based on the sampling time, flow rate, and makeup gas flow rate utilized during each experiment ($k_{avg.\ dilution}$ = $1.4(\pm0.1)\times10^{-5}\ s^{-1}$ during the experiments, and $4.6(\pm0.5)\times10^{-5}\ s^{-1}$ during sampling)."***

**Additionally, Table 1 has been updated with dilution and wall-loss corrected yields. Figure 1 has been updated to report wall-loss corrected hydroxy nitrate (CIMS determined) yields. Figures 3 and 4 have been replaced with wall-loss corrected yield values as well.**

Minor comments:

Additionally, please address the minor comments below. Line numbers are from the track changes version of the manuscript.

Line 60: remove "in forests" as the Rollins et al study was not in a forest.

**This has been removed from line 59 in the revised version of the manuscript.**

Line 319: change to "discussed in Rindelaub et al. (2015)."

**This has been changed in line 306 of the revised manuscript.**

Figure 2: I suggest replacing (A) with Compound A and likewise with (B) to increase visibility of the labels.

**Thank you for this suggestion. This figure has been replaced with the changes suggested.**

Figure 3: Please improve image quality particularly for the equations. The text in light green is particularly difficult to read.

**Thank you for this observation. This figure has been replaced with a higher quality image and the font size has been increased for each line label. The yields reported in this figure have also been updated.**

Figure 8: Image quality could also be improved here.

**This figure has been replaced with a higher quality image.**